EMBO
Molecular Medicine

# Modeling genetic epileptic encephalopathies using brain organoids

Daniel J Steinberg[1] ID, Srinivasarao Repudi[1], Afifa Saleem[2,3], Irina Kustanovich[4], Sergey Viukov[5], Baraa Abudiab[1], Ehud Banne[6,7], Muhammad Mahajnah[8,9], Jacob H Hanna[5] ID, Shani Stern[4], Peter L Carlen[2,3,10] & Rami I Aqeilan[1,*] ID

## Abstract

Developmental and epileptic encephalopathies (DEE) are a group of disorders associated with intractable seizures, brain development, and functional abnormalities, and in some cases, premature death. Pathogenic human germline biallelic mutations in tumor suppressor WW domain-containing oxidoreductase (*WWOX*) are associated with a relatively mild autosomal recessive spinocerebellar ataxia-12 (SCAR12) and a more severe early infantile *WWOX*-related epileptic encephalopathy (WOREE). In this study, we generated an *in vitro* model for DEEs, using the devastating WOREE syndrome as a prototype, by establishing brain organoids from CRISPR-engineered human ES cells and from patient-derived iPSCs. Using these models, we discovered dramatic cellular and molecular CNS abnormalities, including neural population changes, cortical differentiation malfunctions, and Wnt pathway and DNA damage response impairment. Furthermore, we provide a proof of concept that ectopic *WWOX* expression could potentially rescue these phenotypes. Our findings underscore the utility of modeling childhood epileptic encephalopathies using brain organoids and their use as a unique platform to test possible therapeutic intervention strategies.

**Keywords** cerebral organoids; DNA damage; SCAR12; Wnt pathway; WOREE syndrome

**Subject Categories** Genetics, Gene Therapy & Genetic Disease; Neuroscience

## Introduction

Epilepsy is a neurological disorder characterized by a chronic predisposition for the development of recurrent seizures (Fisher *et al*, 2014; Aaberg *et al*, 2017). Epilepsy affects around 50 million people worldwide and is considered the most frequent chronic neurological condition in children (Aaberg *et al*, 2017; Blumcke *et al*, 2017). Approximately 40% of seizures in the early years of life are accounted for by developmental and epileptic encephalopathy (DEE), previously known as early infantile epileptic encephalopathies (EIEEs) (Howell *et al*, 2021). These are pathologies of the developing brain, characterized by intractable epileptiform activity and impaired cerebral and cognitive functions (Lado *et al*, 2013; Shao & Stafstrom, 2016; Nashabat *et al*, 2019; Howell *et al*, 2021). Several genes have been implicated in causing DEEs (McTague *et al*, 2016). In recent years, autosomal recessive mutations in the WW domain-containing oxidoreductase (*WWOX*) gene are increasingly recognized for their role in the pathogenesis of DEE (Piard *et al*, 2018; Nashabat *et al*, 2019). *WWOX*, a tumor suppressor that spans the chromosomal fragile site FRA16D, is highly expressed in the brain, suggesting an important role in central nervous system (CNS) homeostasis (Abu-Remaileh *et al*, 2015). In 2014, *WWOX* was implicated in the autosomal recessive spinocerebellar ataxia-12 (SCAR12) (Gribaa *et al*, 2007; Mallaret *et al*, 2014) and in the *WWOX*-related epileptic encephalopathy (WOREE syndrome, also termed DEE28) (Abdel-Salam *et al*, 2014; Ben-Salem *et al*, 2015; Mignot *et al*, 2015). Both disorders are associated with a wide variety of neurological symptoms, including seizures, intellectual disability, growth retardation, and spasticity, but differ by severity, onset, and underlying types of mutations. The WOREE syndrome is considered more aggressive, appearing as early as 1.5 months and associating with more extreme genetic changes (Banne *et al*, 2021). This observation

1   The Concern Foundation Laboratories, Department of Immunology and Cancer Research-IMRIC, The Lautenberg Center for Immunology and Cancer Research, Hebrew University-Hadassah Medical School, Jerusalem, Israel
2   Biomedical Engineering, University of Toronto, Toronto, ON, Canada
3   Krembil Research Institute, University Health Network, Toronto, ON, Canada
4   Sagol Department of Neurobiology, University of Haifa, Haifa, Israel
5   Department of Molecular Genetics, Weizmann Institute of Science, Rehovot, Israel
6   Genetics Institute, Kaplan Medical Center, Hebrew University-Hadassah Medical School, Rehovot, Israel
7   The Rina Mor Genetic Institute, Wolfson Medical Center, Holon, Israel
8   Paediatric Neurology and Child Developmental Center, Hillel Yaffe Medical Center, Hadera, Israel
9   Rappaport Faculty of Medicine, The Technion, Haifa, Israel
10  Departments of Medicine and Physiology, University of Toronto, Toronto, ON, Canada
    *Corresponding author (lead contact). Tel: +972 2 6758609; E-mail: ramiaq@mail.huji.ac.il

may imply that both syndromes can be considered as a continuum. Alongside seizures, patients with WOREE syndrome may present with global developmental delay, progressive microcephaly, atrophy of specific CNS components, and premature death. However, it is important to note that the phenotypic spectrum of WOREE syndrome is wide, with different patients exhibiting different symptoms. For example, although microcephaly is seen in some patients, many other do not exhibit this condition (Piard *et al*, 2018).

Although modeling WWOX loss of function in rodents has shed some lights on the roles of WWOX in the mammalian brain (Aqeilan *et al*, 2007, 2008; Suzuki *et al*, 2009; Mallaret *et al*, 2014; Tanna & Aqeilan, 2018; Tochigi *et al*, 2019), the genetic background and brain development of a specific patient cannot be modeled in a mouse, but is inherent and retained in patient-derived induced pluripotent stem cells (iPSCs). In an effort to bypass the comprehensible lack of availability of DEE brain samples, including those of patients with *WWOX* mutations, we utilized genome editing and reprogramming technologies to recapitulate the genetic changes seen in patients with WOREE and SCAR12 syndromes in human PSCs. We then generated brain organoids, 3D neuronal cultures, that recapitulate much of the brain's spatial organization and cell type formation, and have neuronal functionality *in vitro* (Amin & Paşca, 2018; Sidhaye & Knoblich, 2020). This allowed us to model features of the development and maturation of the CNS and its complex circuitry, in a system that is more representative of the *in vivo* human physiology than 2D cell cultures. Using this platform, we identified severe defects in neural cell populations, cortical formation, and electrical activity, and tested possible rescue strategies. This approach has resulted in a deeper understanding of WWOX physiology and pathophysiology in the CNS, laying the foundation for developing more appropriate treatments, and supports the concept of using human brain organoids for modeling other human epileptic diseases.

# Results

### Generation and characterization of *WWOX* knockout cerebral organoids

To shed light on the pathogenesis of DEE, we studied the WOREE syndrome as a prototype model using brain organoids. The role of WWOX in the development of the human brain in a controlled genetic background was investigated by generating *WWOX* knockout (KO) clones of the WiBR3 hESC line using the CRISPR/Cas9 system (Abdeen *et al*, 2018). Immunoblot analysis was used to assess WWOX expression in these lines (Fig EV1A). Two clones that showed consistent undetectable protein levels of WWOX throughout our validations were picked for the continuation of the study (Fig EV1A and B)—WWOX-KO line 1B (WKO-1B, from here on KO1) and WKO-A2 (from here on KO2). These clones were assessed for genetic stability and pluripotency using karyotype analysis and teratoma assays, respectively (Appendix Fig S1A and B). Sanger sequencing confirmed editing of *WWOX* at exon 1 (Fig EV1C and Appendix Fig S1C). Furthermore, to confirm cell-autonomous function of WWOX, we restored *WWOX* cDNA into the endogenous *AAVS* locus of WWOX-KO1 hESC line and examined reversibility of the phenotypes (see the Materials and Methods Section). The KO1-

AAV4 line was selected for generating COs for having a strong and stable expression of WWOX throughout our validations (Fig EV1D), and from here on is referred as W-AAV. These lines were practically indistinguishable from the parental cell line (WiBR3 WT) in terms of morphology and proliferation throughout the culture period.

To investigate how depletion of WWOX affects cerebral development in a 3D context, we differentiated our hESCs into cerebral organoids (COs), using an established protocol (Lancaster *et al*, 2013; Lancaster & Knoblich, 2014). COs from all genotypes showed comparable gross morphology and development at all stages. Next, we investigated the expression pattern of WWOX in the developing brain at different time points by co-staining with markers of the two major populations found in the organoids—neuronal progenitor cells and neurons. As seen in week 10, WWOX expression was specifically localized to the ventricular-like zone (VZ), which is composed of SOX2$^+$ cells, corresponding to radial glial cells (RGs), the progenitors of the brain, and not in the surrounding cells (Fig 1A). This finding is in concordance with previous work showing limited WWOX expression during early steps of mouse cortical development (Chen *et al*, 2004). To confirm the identity of the WWOX-expressing cells in the VZ, we co-stained for crystallin alpha B (CRYAB), which is specifically expressed in ventricular radial glial cells (vRGs) (Pollen *et al*, 2015), and confirmed WWOX expression in these cells (Fig 1B). Furthermore, even in later time points, such as week 24, when the VZ structure is lost, WWOX expression is found mainly in SOX2$^+$ cells (Fig EV1E). Importantly, WWOX expression was not seen in COs generated from the WWOX-KO lines, although similar levels of expression of the other markers such as SOX2 and neuron-specific class III β-tubulin (TUBB3 or TUJ1) were observed (Figs 1A and B, and EV1B and E). Interestingly, the W-AAV COs, in which WWOX expression is driven by human ubiquitin promoter (UBP), exhibited high WWOX levels in the VZ, as expected, though other cellular populations also showed prominent WWOX expression (Fig 1A).

Next, to address the microcephalic phenotype observed in some patients, we measured the diameter of our organoids through the culture period, which showed no significant difference (Appendix Fig S2A). This led us to examine the development of the histological cerebral structures. In WT organoids, the VZ, which is composed of SOX2$^+$ cells, is surrounded by intermediate progenitors (IP; TBR2$^+$ cells, also known as EOEMS), marking the presence of the subventricular zone (SVZ). Outside this layer is the cortical plate (CP), composed mainly of neurons (NeuN$^+$ cells) (Appendix Fig S2B). In week 10 COs, no visible differences in the composition or formation of the VZ and the surrounding structures were observed (Fig 1C and Appendix Fig S2C), suggesting similar proportions of these populations. This was also supported by measuring RNA expression levels of progenitor markers (SOX2 and PAX6) and the neuronal marker *TUBB3* (Fig 1D). This surprising observation led us to further examine the two major neuronal subpopulations found in COs—glutamatergic (marked by vesicular glutamate transporter 1, VGLUT1) and GABAergic neurons (marked by glutamic acid decarboxylase 67, GAD67). Immunostaining revealed that although VGLUT1 expression remained similar, a marked increase in expression of GAD67 was observed in KO COs compared with WT. In contrast, WWOX restoration (W-AAV) significantly reversed this imbalance (Fig 1E and F). RNA levels of *SLC17A6 (VGLUT2), SLC17A7 (VGLUT1), GAD1 (GAD67)* and *GAD2 (GAD65)* followed the same trend (Fig 1D).

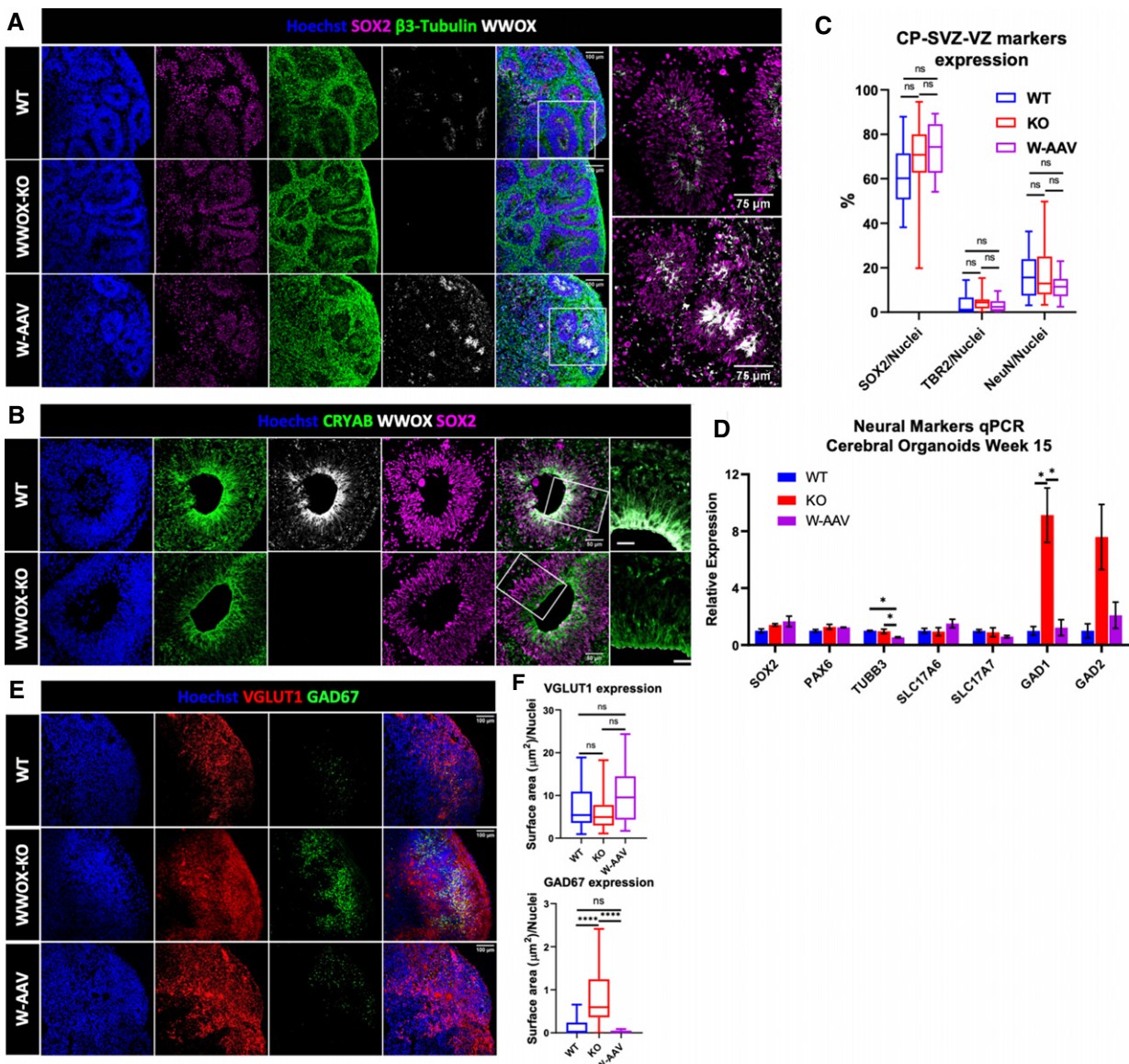

**Figure 1. Generation and characterization of WWOX knockout cerebral organoids.**

A  Week 10 cerebral organoids (COs) stained for the progenitor marker SOX2, neuronal marker β3-tubulin, and WWOX (WT: *n* = 8 from 3 batches, KO: *n* = 8 from 3 batches, and W-AAV *n* = 4 from 1 batch). Scale = 100 μm (left), 75 μm (right).

B  Week 10 COs stained for the ventricular radial glial (vRG) marker CRYAB, together with WWOX and the pan-radial glia marker SOX2. The images on the right are enlargements of the boxed area. The dashed line defines the ventricular space (WT: *n* = 8 from 3 batches, KO: *n* = 8 from 3 batches). Scale = 50 μm (left), 25 μm (right).

C  Quantification of markers that represent the different populations that compose the ventricular-like zone (VZ), subventricular zone (SVZ), and the cortical plate (CP), as depicted in Appendix Fig S2B and C. NeuN marks the mature neurons, TBR2 marks the intermediate progenitors (IPs), and SOX2 marks the ventricular radial glia (vRGs). The boxplot represents the 1st and 3rd quartile, with its whiskers showing the minimum and maximum points and a central band representing the median. Statistical significance was determined using one-way ANOVA with Tukey's multiple comparisons test (WT: *n* = 8 from 3 batches; KO: *n* = 8 from 3 batches; W-AAV: *n* = 4 from 1 batch).

D  qPCR analysis for the assessment of expression levels of different neural markers in 15 weeks COs: SOX2 and PAX6 (progenitor cells), TUBB3 (pan-neuronal), SLC17A6 and SLC17A7 (VGLUT2 and VGLUT1; glutamatergic neurons), and GAD1 and GAD2 (GAD67 and GAD65; GABAergic neurons). The y-axis indicates relative expression fold change. Data are represented as mean ± SEM. Statistical significance was determined using one-way ANOVA with Tukey's multiple comparisons test (WT: *n* = 4 from 1 batch; KO: *n* = 4 from 1 batch; and W-AAV: *n* = 4 from 1 batch).

E  Immunofluorescent (IF) staining for the glutamatergic neuron marker VGLUT1 and GABAergic neuron marker GAD67 (GAD1) in week 10 COs (WT: *n* = 8 from 3 batches; KO: *n* = 8 from 3 batches; and W-AAV: *n* = 4 from 1 batch). Scale = 100 μm.

F  Quantification of the images represented in Fig 1E. VGLUT1 and GAD67 were quantified as the surface area covered by the staining and were normalized to the number of nuclei in each image. The boxplot represents the 1st and 3rd quartile, with its whiskers showing the minimum and maximum points and a central band representing the median. As the samples were not normally distributed, statistical significance was determined using the Kruskal–Wallis test with Dunn's multiple comparisons test. Outliers were removed using the ROUT test (Q = 1%).

Data information: *n.s* (non-significant), **P* ≤ 0.05, and *****P* ≤ 0.0001.

These findings suggest that during human embryonic development, WWOX expression is limited to the cells of the apical layer of the VZ, and that WWOX depletion did not affect the VZ-SVZ-CP architecture but did disrupt the balance between glutamatergic and GABAergic neurons.

### WWOX-depleted cerebral organoids exhibited hyperexcitability and epileptiform activity

Cerebral organoids give rise to neurons that have previously shown electrophysiological functionality (Trujillo *et al*, 2019). To characterize the functional properties of the WWOX-KO COs, we performed local field potential recordings (LFP) on 7-week CO slices. Electrodes were positioned 150 μm away from the edge of the slice (Fig EV1F) to avoid areas potentially damaged by slice preparation. Sample traces of the WT and KO COs revealed visible differences between the two lines under baseline conditions (Fig 2A, left). The mean spectral power of field recordings further showed an overall increase in power of the KO COs in the 0.25–1 Hz, typically labelled as slow-wave oscillations (SWO, < 1 Hz) (Fig 2B), and a decrease in the 30- to 79.9-Hz high-frequency range (Fig EV1G and H). The oscillatory power (OP) was quantified by the area under the curve, which was significantly higher than the WT line, under baseline conditions (Fig 2C). Over time, the OP of the KO lines decreased significantly, while the WT line's OP stayed the same (Fig EV1I), suggesting a developmental delay in the KO line.

To further measure the hyperexcitability of the KO line, 100 μM 4-AP, a commonly used convulsant for seizure induction, was applied to the slices during recordings. While 4-AP did show changes in LFP recordings for both WT and KO lines (Fig 2A, right), the KO line showed significantly increased activity, which was otherwise absent in WT traces (Fig EV1J). The effect of 4-AP on spectral power became evident 5 min after its addition, as indicated by the sample spectrogram (Fig 2E). Cross-frequency coupling of the sample traces for both WT and KO lines in the presence of 4-AP revealed an increase in the δ: HFO frequency pairs—an attribute that has previously been used to characterize and classify seizure substates (Fig 2D) (Guirgis *et al*, 2013). Importantly,

transduction of a lentivirus containing *WWOX* cDNA resulted in recovery of the KO line, with respect to the mean power spectral density (Fig 2F and G).

### WWOX-depleted cerebral organoids exhibited impaired astrogenesis and DNA damage response

It is widely accepted that an imbalance between excitatory and inhibitory activity in the brain is a leading mechanism for seizures, but this does not necessarily mean neurons are the only population involved. It is well known that brain samples from epileptic patients show signs of inflammation, astrocytic activation, and gliosis (Cohen-Gadol *et al*, 2004; Thom, 2009), which can be a sole histopathological finding in some instances (Blumcke *et al*, 2017). Whether this phenomenon is a result of the acute insult or a cause of the seizures is still debatable (Vezzani *et al*, 2011; Robel *et al*, 2015; Rossini *et al*, 2017; Patel *et al*, 2019). Furthermore, recent work has demonstrated astrogliosis in the brain of *Wwox*-null mice (Hussain *et al*, 2019).

To address this, we used immunofluorescence staining to visualize the astrocytic markers glial fibrillary acidic protein (GFAP) and S100 calcium-binding protein B (S100β) in week 15 and week 24 COs (Fig 3A–C). This revealed a marked increase in astrocytic cells in WWOX-KO COs that progressed through time, and was partially reversed in W-AAV COs. This was further supported by immunoblot analysis of week 20 COs (Fig EV2A and B). It is notable that GFAP also marks RGs (Middeldorp *et al*, 2010), which are abundant in week 15, but are reduced in number in week 24, which could be a source of noise at early stages.

Astrocytes arise from two distinct populations of cells in the brain: the RG cells, switching from neurogenesis to astrogenesis, or the astrocyte progenitor cells (APCs) (Zhang *et al*, 2016; Blair *et al*, 2018). To track back these differences in astrocyte markers, we compared 6- and 10-week-old COs. We found that in week 6 COs, where no astrocytic markers were detected in WT organoids, a significant expression of S100β was observed in the VZ (Fig 3D). At week 10, we observed expression of S100β in both WT and KO organoids, but when co-staining is performed with the cell proliferation

---

**Figure 2.  WWOX-KO cerebral organoids demonstrated hyperexcitability and epileptiform activity.**

Sample recordings from 7-week-old hESC-derived cerebral organoids (COs).

A    Sample traces show visible differences in local field potential, with WWOX-KO COs (red) showing increased activity compared with WT (blue) in baseline condition (left) and in the presence of 100 μM 4-AP (right).

B    Mean spectral power of week 7 WT and KO COs in baseline conditions. The line marks 0.25- to 1-Hz frequency range. Statistical significance was determined using the two-tailed unpaired Welch's *t*-test (WT: *n* = 14 slices, 5 organoids, and 3 batches; KO: *n* = 14 slices, 8 organoids, and 3 batches).

C    Normalized area under the curve of the mean spectral power in (B) for the 0.25- to 1-Hz frequency range. Data represented by mean ± SEM. Statistical significance was determined using the two-tailed unpaired Welch's *t*-test. The numerals in all bars indicate the number of analyzed slices and organoids (i.e., slices (organoids)).

D    CFC analysis shows increased coupling in the δ: HFO frequency pairs. The color scale represents the modulation index—a measure of the coupling between the phase of low-frequency oscillations and amplitude of high-frequency oscillations in a given signal.

E    Sample spectrogram of WWOX-KO slice shows a gradual increase in activity upon addition of 4-AP (marked by red arrow) for up to 2 min, and a decrease after 25 min. All traces were filtered with a 60 Hz notch filter and 0.5 Hz high-pass filter.

F, G   WWOX's coding sequence was reintroduced into week 6 WWOX-KO COs using lentiviral transduction (lenti-WWOX) (see Materials and Methods). (F) Immunofluorescent staining showing WWOX expression in different populations in WWOX-KO organoids following infection with lentivirus. NT = non-treated. Scale = 50 μm. (G) Normalized area under the curve of the mean spectral power of WT line, 2 KO lines, and 2 KO lines infected with lenti-WWOX at week 7 in baseline condition, for the 0.25- to 1-Hz frequency range. Data represented by mean ± SEM. The numerals in all bars indicate the number of analyzed slices and organoids (i.e., slices (organoids)). Statistical significance was determined using one-way ANOVA with Tukey's multiple comparisons test.

Data information: ***$P \leq 0.001$ and ****$P \leq 0.0001$.

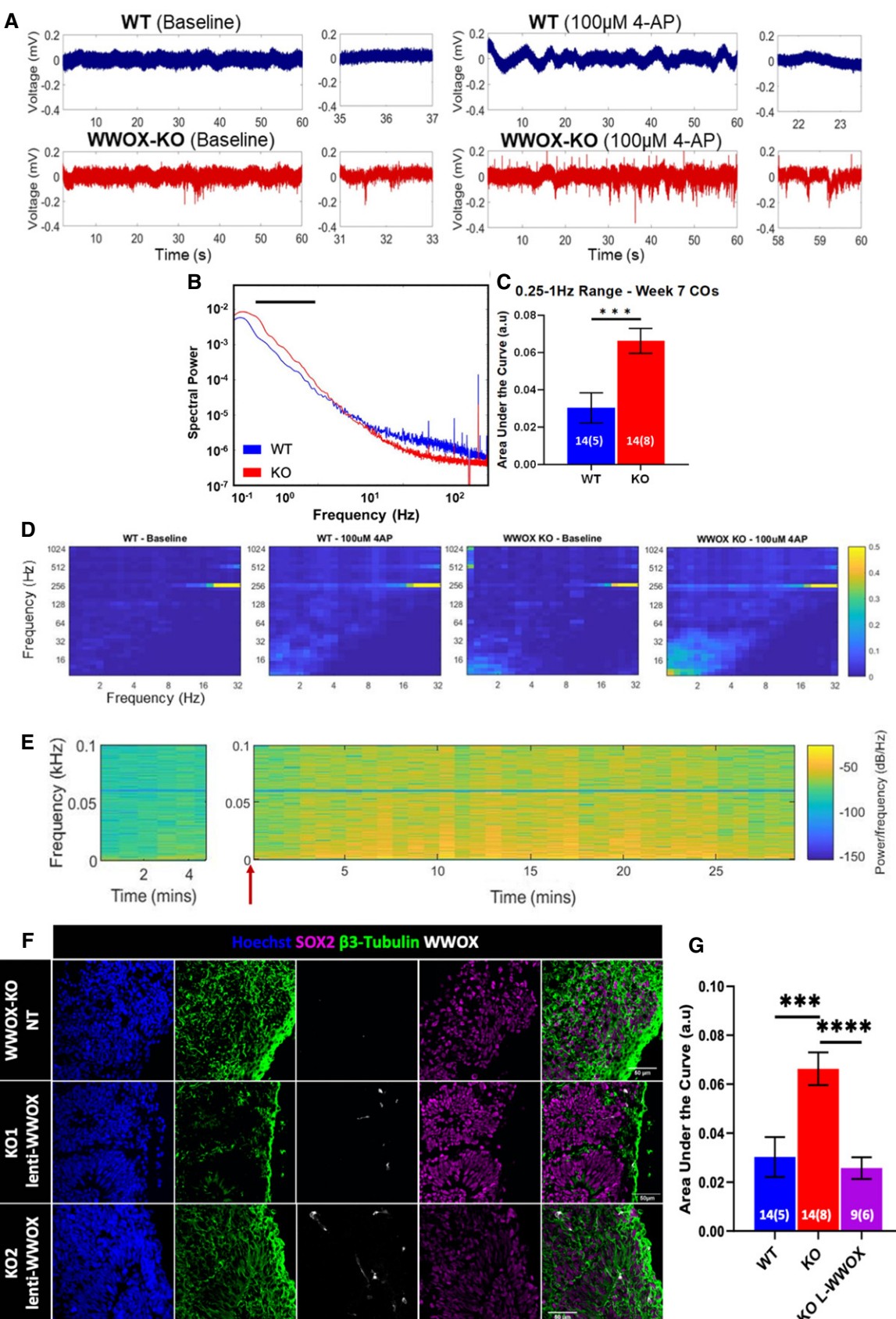

**Figure 2.**

marker Ki67, we did not detect a significant difference in double-positive cells suggesting similar glial proliferation (Fig EV2C). Quantification of Ki67$^+$ nuclei, together with SOX2$^+$ nuclei, revealed that although the proportions of SOX2 remained intact in WT compared with WWOX-KO (18.5% in WT, 95% CI = 14.2–22.81; 19.5% in KO, 95% CI = 15.29–22.81), the proportions of proliferating cells (9.5% in WT, 95% CI = 6.63–12.35; 4.9% in KO, 95% CI = 2.76–7.13) and Ki67$^+$/SOX2$^+$ double-positive cells were diminished (51.83% in WT, 95% CI = 41.18–62.48; 27.09% in KO, 95% CI = 14.1–40.1) (Fig EV2D). These findings imply that although overall proliferation of SOX2$^+$ cells is decreased upon loss of WWOX, the total amount of RGs outside the VZ is not affected, raising a question in regard to the source of the astrocytic cells.

This peculiar behavior of the vRGs in KO COs led us to take a closer look at their functionality by examining their physiological DNA damage response (DDR), a signaling pathway in which WWOX is known to be directly involved (Abu-Odeh *et al*, 2014b; Abu-odeh *et al*, 2016). To this end, we stained for γH2AX and 53BP1, surrogate markers for DNA double-strand breaks. We found a marked accumulation of γH2AX and 53BP1 foci in the nuclei of SOX2$^+$ cells in the innermost layer of the VZ, averaging 1.5 foci/nuclei [95% CI =1.33–1.74] and 1.2 foci/nuclei [95% CI = 1–1.38] in WWOX-KO, respectively. This is in comparison with 0.78 γH2AX foci/nuclei [95% CI = 0.55–1.02] and 0.62 53BP1 foci/nuclei in the age-matched WT COs, and with 0.58 foci/nuclei [95% CI = 0.38–0.77] and 0.56 foci/nuclei [95% CI = 0.37–0.76] in the age-matched W-AAV COs (Fig 3E and F). These findings are consistent with WWOX direct role in DDR signaling (Aqeilan *et al*, 2014; Hazan *et al*, 2016). Importantly, W-AAV COs presented with improved DDR. Intriguingly, a higher number of γH2AX foci were found in highly proliferating cells in the VZ, which was observed by co-staining with Ki67 —18.6% of SOX2$^+$ cells were double-positive in KO COs [95% CI = 15–22%] compared with 11.9% [95% CI = 8–15%] (Fig EV2E and F). This led us to speculate that the continued proliferation of damaged cells might be accompanied by diminished apoptosis, which might indicate loss of checkpoint inhibition. This hypothesis was addressed by staining for cleaved caspase-3 in the VZ, which revealed a decline in apoptosis of these cells upon WWOX-KO, and was rescued in the W-AAV COs (Fig EV2G and H).

In conclusion, WWOX-KO COs present with a progressive increase in astrocytic number, likely due to enhanced differentiating RGs, and with increased DNA damage in neural progenitor cells.

## RNA-sequencing of WWOX-depleted cerebral organoids revealed major differentiation defects

In an effort to examine the molecular profiles, we performed whole-transcriptome RNA-sequencing (RNA-seq) analysis on week 15 WT and KO COs. Albeit the known heterogeneity of brain organoids, principal component analysis (PCA) separated the sample into two distinct clusters (Fig EV3A). The analysis revealed 15,370 differentially expressed genes, of which 1,246 genes were upregulated in WWOX-KO COs and showed both a fold change greater than 1.2 (FC > 1.2) and a significant *P*-value (*P* < 0.01), and 1,021 genes were downregulated (FC < 1/1.2, *P* < 0.01) (Fig EV3B; Tables EV1 and EV2). Among the top 100 upregulated genes, we found genes related to neural populations such as GABAergic neurons (*GAD1, GRM7, LHX5*) and astrocytes (*AGT, S100A1, GJA1, OTX2*), and to neuronal processes such as calcium signaling (*HRC, GRIN2A, ERBB3, P2RX3, HTR2C, PDGFRA*) and axon guidance (*GATA3, DRGX, ATOH1, NTN1, SHH, RELN, OTX2, SLIT3, GBX2, LHX5*) (Fig 4A). In the top 100 downregulated genes were genes related to GABA receptors (*GABRB3, GABRB2*), autophagy (*IFI16, MDM2, RB1, PLAT, RB1CC1*), and the mTOR pathway (*EIF4EBP1, PIK3CA, RB1CC1*) (Fig 4A).

Gene set enrichment analysis (GSEA) and gene ontology (GO) enrichment analysis of the top 3,000 differentially expressed genes revealed, among others, inhibition of processes related to ATP synthesis-coupled electron transport and oxidative phosphorylation (Appendix Fig S3A), all of which are consistent with previous reported functions of WWOX in mouse models (Abu-Remaileh & Aqeilan, 2014, 2015; Abu-Remaileh *et al*, 2018). Downregulation of genes related to negative regulation of cell cycle was seen (Fig 4B), consistent with the previously reported diminished checkpoint inhibition (Abu-Odeh *et al*, 2014b; Abu-odeh *et al*, 2016). On the other hand, marked enrichment was seen in pathways related to regionalization, neuron fate commitment and specification, axis specification (ventral–dorsal and anterior–posterior), and glycolysis and

---

**Figure 3.  WWOX-KO cerebral organoids showed impaired astrogenesis and DNA damage response.**

A   Week 15 and week 24 COs stained for the astrocytic and radial glial marker GFAP, and the astrocyte-specific marker S100β (WT W15: *n* = 9 from 3 batches; KO W15: *n* = 16 from 3 batches; W-AAV W15: *n* = 4 organoids from 1 batch; WT W24: *n* = 10 from 4 batches KO W24: *n* = 9 from 3 batches; and W-AAV W24: *n* = 4 from 1 batch). Scale = 100 μm.

B   qPCR analysis of astrocytic markers in COs at week 15. The y-axis indicates relative expression fold change. Data are represented as mean ± SEM. Statistical significance was determined using one-way ANOVA with Tukey's multiple comparisons test (WT: *n* = 4 from 1 batch; KO: *n* = 4 from 1 batch; and W-AAV: *n* = 4 from 1 batch).

C   qPCR analysis of astrocytic markers in COs at week 24. The y-axis indicates relative expression fold change. Data are represented as mean ± SEM. Statistical significance was determined using one-way ANOVA with Tukey's multiple comparisons test (WT: *n* = 4 from 2 batches; KO: *n* = 3 from 2 batches; and W-AAV *n* = 3 from 1 batch).

D   IF staining of week 6 COs for astrocytic markers in the surrounding of VZs (WT: *n* = 6 from 2 batches, and KO: *n* = 6 from 2 batches). Scale = 100 μm (left) and 50 μm (right).

E   Staining for the DNA damage markers γH2AX and 53BP1 in the nuclei of cells in the VZ of week 6 COs at physiological conditions, together with the pan-radial glial marker SOX2 (WT: *n* = 8 from 3 individual batches; KO: *n* = 12 from 3 individual batches; and W-AAV: *n* = 4 from 1 batch). Scale = 50 μm (left), 25 μm (right).

F   Quantification of γH2AX foci (top) and 53BP1 foci (bottom) in the nuclei of cells composing the innermost layer of the VZ, normalized to the total number of nuclei in this layer. The boxplot represents the 1$^{st}$ and 3$^{rd}$ quartile, with its whiskers showing the minimum and maximum points and a central band representing the median. Statistical significance was determined using one-way ANOVA with Tukey's multiple comparisons test (WT: *n* = 8 organoids from 3 batches; KO: *n* = 12 organoids from 3 batches; and W-AAV: *n* = 4 from 1 batch).

Data information: *n.s* (non-significant), *$P \leq 0.05$, **$P \leq 0.01$, and ****$P \leq 0.0001$.

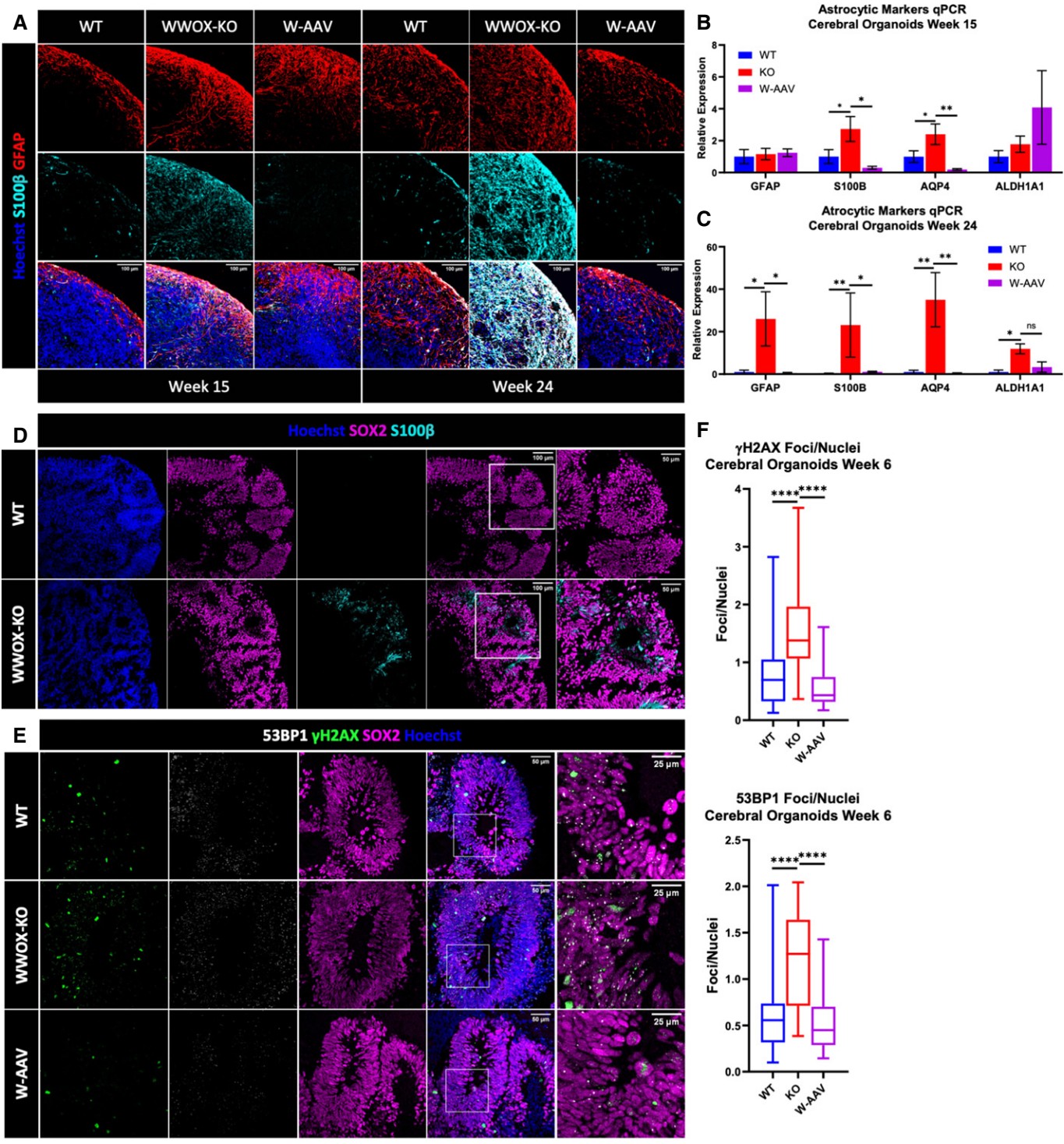

**Figure 3.**

gluconeogenesis (Fig 4B and Appendix Fig S3A and B), some of which are also supported by past studies (Wang *et al*, 2012; Abu-Remaileh & Aqeilan, 2014). As could be anticipated, upregulated genes were related to the development pathways such as Wnt pathway *(e.g., WNT1, WNT2B, WNT3, WNT3A, WNT5A, WNT8B, LEF1,*

*AXIN2, GBX2, ROR2, LRP4, NKD1, IRX3, CDH1)* and the Shh pathway *(e.g., SHH, GLI1, LRP2, PTCH1, HHIP, PAX1, PAX2)*.

Since WWOX has been previously implicated in the Wnt signaling pathway (Bouteille *et al*, 2009; Wang *et al*, 2012; Abu-Odeh *et al*, 2014a; Cheng *et al*, 2020; Khawaled *et al*, 2020), we set to

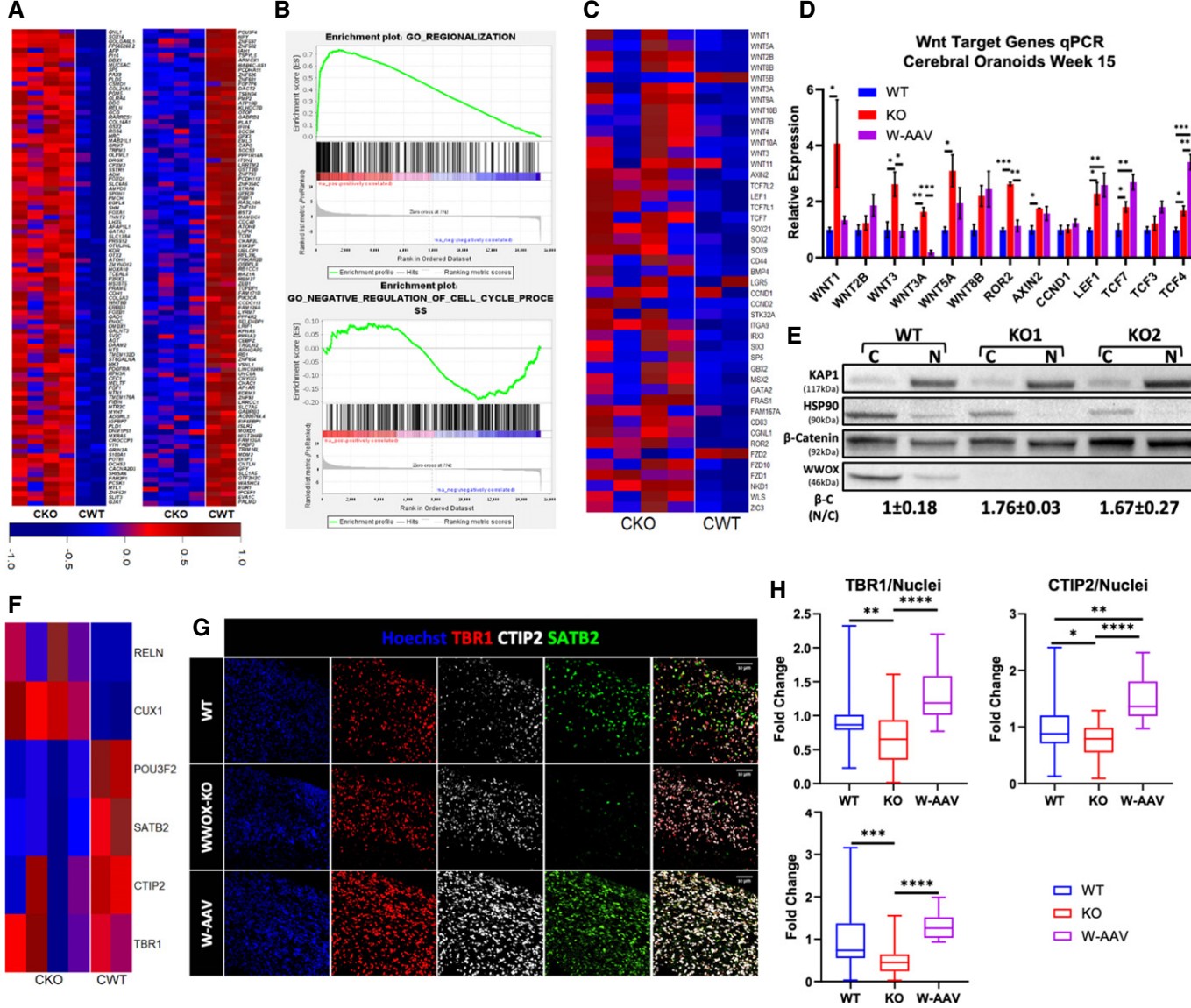

**Figure 4. Cerebral organoid RNA sequencing revealed major differentiation defects.**

RNA sequencing (RNA-seq) of week 15 COs and transcriptome analysis (WT: $n = 2$, KO: $n = 4$).

A　Heatmap of 100 upregulated genes (left panel) and 100 downregulated genes (right panel) selected by highest fold change.

B　Gene set enrichment analysis (GSEA) revealed enrichment of genes related to regionalization of the organoids, and decreased expression of genes related to negative regulation of the cell cycle.

C　Heatmap of Wnt pathway-related genes in 15-week COs.

D　qPCR analysis for selected Wnt target genes validating the results of the RNA-seq. The y-axis indicates relative expression fold change. Data are represented as mean ± SEM. Statistical significance was determined using one-way ANOVA with Tukey's multiple comparisons test (WT: $n = 4$ from 1 batch; KO: $n = 4$ from 1 batch; and W-AAV: $n = 4$ from 1 batch).

E　Week 16 COs were subfractionated into a cytoplasmic (C) and nuclear (N) fractions. The experiment was run twice with a total of 2 WT organoids and 4 KOs organoids (2 for each KO line). KAP-1 marks the nucleus, and HSP90 marks the cytoplasm. The numbers at the bottom are a quantification of the nuclear fraction of β-catenin band intensities normalized to the cytosolic fraction [β-C (N/C)].

F　Heatmap showing the expression levels of markers of the six different layers of the human cortex in week 15 organoids from deepest to the most superficial: TBR1, BCL11B (CTIP2), SATB2, POU3F2 (BRN2), CUX1, and RELN.

G　IF staining in week 15 COs validating the decreased levels of the deep layer cortical markers CTIP2 (BCL11B) and TBR1, and superficial layer marker SATB2 (WT: $n = 3$; KO: $n = 4$; and W-AAV: $n = 4$). Scale = 50 μm.

H　Quantification of the cortical markers seen in G, normalized to the total number of nuclei. The y-axis indicates the fold change compared with the average of the WT COs. The boxplot represents the 1st and 3rd quartile, with its whiskers showing the minimum and maximum points and a central band representing the median. Statistical significance was determined using one-way ANOVA with Tukey's multiple comparisons test (WT: $n = 9$ from 3 batches; KO: $n = 16$ from 3 batches; and W-AAV: $n = 4$ organoids from 1 batch).

Data information: *$P \leq 0.05$, **$P \leq 0.01$, ***$P \leq 0.001$, and ****$P \leq 0.0001$.

further explore this in our CO models. First, we used our RNA-seq data to inspect the expression of different members of the WNT signaling pathway (such as *WNT1, WNT3, WNT5A, WNT8B*), of canonical targets (such as *Axin2, TCF7L2, LEF1, TCF7L1*), of brain-specific targets (*IRX3, ITGA9, GATA2, FRAS1, SP5*), and of receptors (*ROR2, FZD2, FZD10, FZD1*) (Fig 4C). We next validated some of these genes using qPCR (Fig 4D). We also observed downregulation of some of the Wnt-related genes in W-AAV COs, including WNT3, WNT3A, WNT1, and ROR2 (Fig 4D). Additionally, to prove Wnt activation, we demonstrated β-catenin translocation into the nucleus—a hallmark of the canonical Wnt pathway. To this end, week 16 COs were subfractionated into cytoplasmic and nuclear fractions and immunoblotted (Fig 4E). We found an approximate 1.7-fold increase in the normalized intensity of β-catenin in the nucleus of WWOX-KO COs, supporting the notion of Wnt pathway activation after the loss of WWOX. This was also supported by examining kinetic expression of several Wnt-related genes (*WNT3, AXIN2, LEF1,* and *TCF7*) at weeks 6–24, which revealed a chronic Wnt activation in KO COs as compared to progressive decrease in WT COs (Fig EV3C).

Recent evidence has demonstrated that activation of Wnt in forebrain organoids during development causes a disruption of neuronal specification and cortical layer formation (Qian *et al*, 2020). To address whether this occurs in WWOX-KO COs, we examined the expression levels of cortical layer markers (Qian *et al*, 2016) in our RNA-seq data (Fig 4F). Interestingly, changes were observed in all six layers, with layers I-IV (marked by *TBR1, BCL11B, SATB2, POU3F2*) showing decreased expression and superficial layers V-VI (marked by CUX1 and RELN) exhibiting marked increase. This pattern was also confirmed by qPCR (Fig EV3D). Intriguingly, when we examined protein levels using immunofluorescent staining, we also observed impaired expression patterns and layering, with TBR1$^+$, CTIP2$^+$ (BCL11B), and SATB2$^+$ neurons intermixing in WWOX-KO COs (Fig 4G and H). This defect was progressive, worsening at week 24 (Fig EV3E and F). Surprisingly, when examining the effect of ectopic WWOX expression, a less clear phenotype was observed; although CTIP2$^+$ cell and SATB2$^+$ cell numbers recovered and layering improved in W-AAV COs compared with WWOX-KO COs, RNA levels did only partially (Fig 4G and H). In contrast, expression of the superficial layer markers CUX1 and RELN, which was upregulated in WWOX-KO, decreased in W-AAV, together with the upper layer marker POU3F2 (BRN2). Importantly, when examining the expression of dorsal and ventral genes in COs, we observed that the organoids are of dorsal identity, as assessed by high RNA read counts (Appendix Fig S3C). Of note, we did not observe any statistically significant differences between WT and KO in the expression of these markers (Appendix Table S1).

Overall, RNA-seq reveled impaired spatial patterning, axis formation, and cortical layering in WWOX-KO COs, which is correlated with disruption of cellular pathways and activation of Wnt signaling. The reintroduction of WWOX prevents these changes to some extent, further supporting its possible implication in gene therapy.

## Brain organoids of patient-derived WWOX-related developmental and epileptic encephalopathies

Although disease modeling using CRISPR-edited cell is a widely used tool, critiques argue against it for not modeling the full genetic background of the human patients. Therefore, we reprogrammed peripheral blood mononuclear cells (PBMCs) donated from two families with WWOX-related diseases, differing in their severity: The first family carries a c.517-2A>G splice site mutation (Weisz-Hubshman *et al*, 2019) that results in the WOREE syndrome (DEE28) phenotype in the homozygous patient (referred to as WSM family) (Appendix Fig S4A and B); and the second family carries a c.1114G>C (G372R) mutation (Mallaret *et al*, 2014) that results in the SCAR12 phenotype in the homozygous patient (referred to as WPM family) (Appendix Fig S6A and B). All the iPSC lines showed normal morphology for primed hPSCs and self-renewal capabilities and were evaluated for expression of pluripotent markers (Appendix Figs S4C and D, and S6C and D).

We then proceeded to generate COs from iPSCs isolated from the healthy, heterozygote parents of the WSM family (the father, referred to as WSM F1; and the mother, referred to as WSM M2, collectively referred to as WSM P), and from the sick homozygote son (lines WSM S2 and S5, referred to collectively as WSM S). Additionally, we employed the rescue approach descried for W-AAV COs and reintroduced *WWOX* into the WSM S5 line (referred to as WSM S5 W-AAV3 and W-AAV6, and collectively as WSM S W-AAV) (Appendix Fig S4A). These organoids were then validated for neuronal differentiation and VZ formation (Fig 5A). Similar to the hESC-derived COs, WWOX was mainly expressed in the VZ in WSM F1 and WSM M2. Although β3-Tubulin$^+$-positive cells and SOX2$^+$-positive cells were comparable in numbers, WSM S COs showed no detectable levels of WWOX, and WSM S W-AAV COs expressed WWOX globally.

Next, to evaluate the neuronal hyperexcitability of the WSM S COs, we performed cell-attached recordings from 41 WSM S COs' neurons along with 24 neurons from WSM P COs and 40 neurons from WSM S W-AAV organoids. Sample traces with the spontaneous firing of action potentials from the WSM S, WSM P, and WSM S W-AAV COs revealed visible differences between the three groups recorded under the same conditions. WSM S COs demonstrated bursts of action potentials and overall elevated neuronal activity compared with the WSM P and WSM S W-AAV organoids (Fig 5B). The WSM S COs' neurons showed a drastic increase in the firing rate (about a fourfold) compared with the WSM P ($P < 0.0001$) and WSM S W-AAV ($P < 0.0001$) COs' neurons (Fig 5C). There was no significant difference between the firing rate of WSM P and WSM S W-AAV COs' neurons ($P = 0.7681$). Importantly, no significant difference was observed when comparing COs from WSM F1 line to WSM M2 line ($P = 0.0952$) (Appendix Fig S4E). Overall, our results demonstrate neuronal hyperexcitability in 7-week-old organoids derived from the patient with WOREE syndrome compared with his or her parents.

Consistent with our previous findings, week 10 WSM S COs exhibited increased expression of GAD67 compared with WSM F1 and WSM M2, a phenotype that was reversed in WSM S W-AAV COs (Fig 5D and E). The expression of VGLUT1 remained unchanged between the different lines. We next assessed the expression of astrocytic markers and found increased expression of GFAP and S100β in WSM S compared with age-matched controls (Fig 6A). DDR defects in vRGs of WSM S COs were also apparent (Fig 6B and C). The status of the Wnt pathway was also assessed using qPCR, with findings suggestive of activation in week 10 WSM S COs compared with the

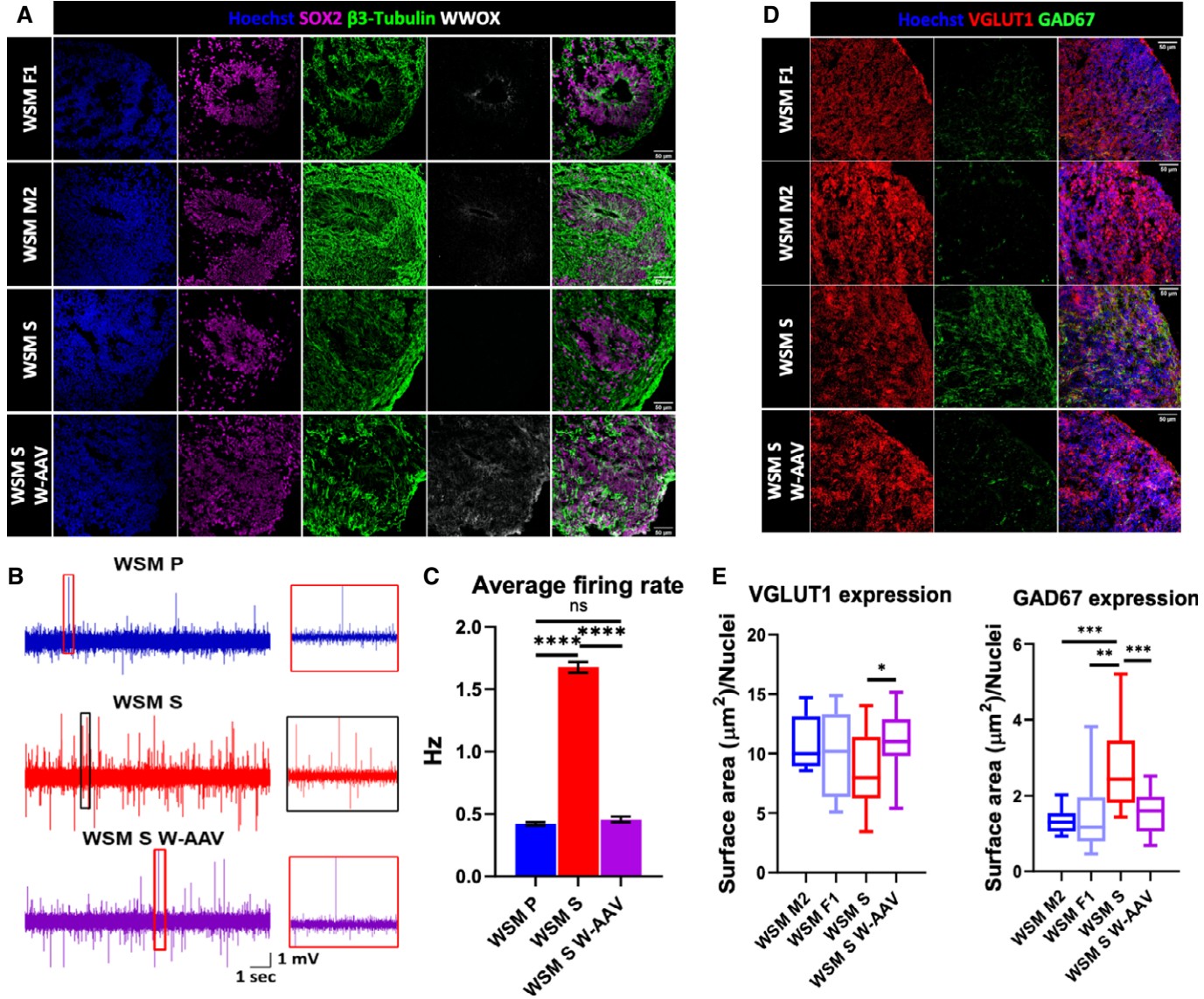

**Figure 5. WWOX-related epileptic encephalopathy cerebral organoids recapitulate neuronal abnormalities.**

Peripheral blood mononuclear cells (PBMCs) were isolated from a patient with WOREE syndrome and from his healthy parents and were reprogrammed into iPSCs, and subsequently were differentiated into COs.

A  Week 10 WSM COs stained for the progenitor marker SOX2, neuronal marker β3-tubulin, and WWOX (WSM F1: $n = 2$ from 1 batch; WSM M2: $n = 2$ from 1 batch; WSM S2: $n = 2$ from 1 batch; WSM S5: $n = 2$ from 1 batch; WSM S5 W-AAV3: $n = 2$ from 1 batch; and WSM S5 W-AAV6: $n = 2$ from 1 batch). Scale = 50 μm.

B  Representative traces of spontaneous spikes recorded from neurons of WSM P (blue), WSM S (red), and WSM S W-AAV (purple) week 7 COs. Each recording is 12 s long, and the zoom-in is of 0.5 s (red box on the right).

C  Average firing rate over 24 neurons from WSM P COs (4 organoids), 41 neurons from WSM S COs (3 organoids), and 40 neurons from WSM S W-AAV organoids (3 organoids). Statistical significance was determined using one-way ANOVA with Tukey's multiple comparisons test. Bars represents the mean ± SEM.

D  GAD67 and VGLUT1 immunostaining in week 10 WSM COs (WSM F1: $n = 2$ from 1 batch; WSM M2: $n = 2$ from 1 batch; WSM S2: $n = 2$ from 1 batch; WSM S5: $n = 2$ from 1 batch; WSM S5 W-AAV3: $n = 2$ from 1 batch; and WSM S5 W-AAV6: $n = 2$ from 1 batch). Scale = 50 μm.

E  Quantification of the data shown in Fig 5D. Statistical significance was determined using one-way ANOVA with Tukey's multiple comparisons test (WSM F1: $n = 2$ from 1 batch; WSM M2: $n = 2$ from 1 batch; WSM S2: $n = 2$ from 1 batch; WSM S5: $n = 2$ from 1 batch; WSM S5 W-AAV3: $n = 2$ from 1 batch; and WSM S5 W-AAV6: $n = 2$ from 1 batch).

Data information: ns (non-significant), *$P \leq 0.05$, **$P \leq 0.01$, ***$P \leq 0.001$, and ****$P \leq 0.0001$.

age-matched WSM P and WSM S W-AAV (Fig 6D). Finally, we evaluated cortical lamination using immunofluorescence and found decreased expression of cortical markers CTIP2[+] and SATB2[+] in WSM S COs (Fig 6E and F).

Since a major part of the phenotype was observed in the cortical areas of the COs, and since a recent paper has demonstrated a major role for *WWOX* in the cortex (Repudi *et al*, 2021), we decided to employ a cortex-specific protocol and generate forebrain organoids

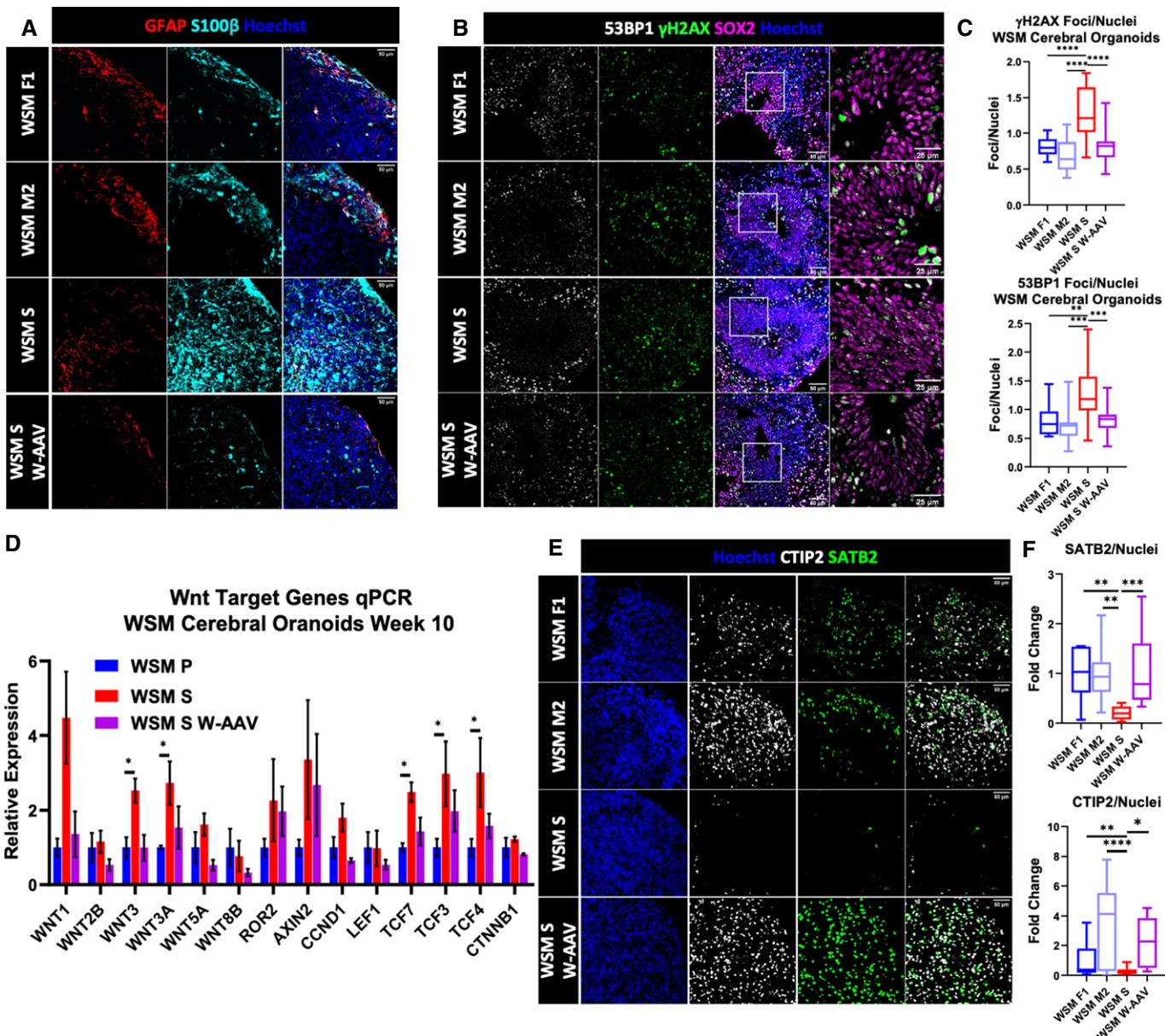

**Figure 6. WWOX-related epileptic encephalopathies depict molecular abnormalities similar to a complete WWOX loss.**

A Week 15 WSM COs stained for astrocytic markers GFAP and S100β. (WSM F1: $n = 2$ from 1 batch; WSM M2: $n = 3$ from 1 batch; WSM S2: $n = 2$ from 1 batch; WSM S5: $n = 2$ from 1 batch; WSM S5 W-AAV3: $n = 2$ from 1 batch; and WSM S5 W-AAV6: $n = 2$ from 1 batch). Scale = 50 μm.

B Impaired DNA damage response in WOREE-derived organoids. SOX2 marks the radial glia in the VZ (WSM F1: $n = 4$ from 1 batch; WSM M2: $n = 4$ from 1 batch; WSM S2: $n = 4$ from 1 batch; WSM S5: $n = 4$ from 1 batch; WSM S5 W-AAV3: $n = 4$ from 1 batch; and WSM S5 W-AAV6: $n = 4$ from 1 batch). Scale = 50 μm (left) and 25 μm (right).

C Quantification of DNA damage foci, marked by γH2AX (left) and 53BP1 (right), in the nuclei of the cells in the VZs of week 6 WSM COs. The boxplot represents the 1st and 3rd quartile, with its whiskers showing the minimum and maximum points and a central band representing the median. Statistical significance was determined using one-way ANOVA with Tukey's multiple comparisons test (WSM F1: $n = 4$ from 1 batch; WSM M2: $n = 4$ from 1 batch; WSM S2: $n = 4$ from 1 batch; WSM S5: $n = 4$ from 1 batch; WSM S5 W-AAV3: $n = 4$ from 1 batch; and WSM S5 W-AAV6: $n = 4$ from 1 batch).

D qPCR analysis for selected Wnt target genes in week 10 WSM COs. The y-axis indicates relative expression fold change. Data are represented as mean ± SEM. Statistical significance was determined using one-way ANOVA with Tukey's multiple comparisons test (WSM P: $n = 6$ from 1 batch; WSM S: $n = 4$ from 1 batch; and WSM S W-AAV: $n = 4$ from 1 batch).

E Week 15 WSM COs stained for the cortical layers' markers CTIP2 and SATB2. (WSM F1: $n = 2$ from 1 batch; WSM M2: $n = 3$ from 1 batch; WSM S2: $n = 2$ from 1 batch; WSM S5: $n = 2$ from 1 batch; WSM S5 W-AAV3: $n = 2$ from 1 batch; and WSM S5 W-AAV6: $n = 2$ from 1 batch). Scale = 50 μm.

F Quantification of the staining presented in E. The boxplot represents the 1st and 3rd quartile, with its whiskers showing the minimum and maximum points and a central band representing the median. Statistical significance was determined using one-way ANOVA with Tukey's multiple comparisons test (WSM F1: $n = 2$ from 1 batch; WSM M2: $n = 3$ from 1 batch; WSM S2: $n = 2$ from 1 batch; WSM S5: $n = 2$ from 1 batch; WSM S5 W-AAV3: $n = 2$ from 1 batch; and WSM S5 W-AAV6: $n = 2$ from 1 batch).

Data information: *$P \leq 0.05$, **$P \leq 0.01$, ***$P \leq 0.001$, and ****$P \leq 0.0001$.

(FOs) (Qian *et al*, 2016, 2018). First, to validate reproducibility, we generated FOs from WSM F1 and WSM S5 and found comparable phenotypes for WSM COs (Appendix Figs S4F and G, and S5).

We next sought to study whether the WPM SCAR12 family, whose patients have a milder phenotype, present with similar phenotypes to our COs and FOs of WOREE syndrome. We generated FOs from the healthy heterozygous father and mother (WPM F2 and WPM M3) and their affected homozygous daughter and son (WPM D1 and WPM S1). As expected, FOs were indistinguishable in terms of morphology, growth, and expression of β3-Tubulin and SOX2 (Fig EV4A), but while in the VZ of WPM F2 and WPM M3, WWOX was detected, barely any signal was observed in WPM D1 and S1, consistent with WWOX levels in the iPSCs (Appendix Fig S6A). Dorsal forebrain identity was validated through staining for PAX6 (Appendix Fig S6E). Surprisingly, transcript expression levels of neuronal markers did not show any clear difference in the ratio between glutamatergic and GABAergic neurons (Fig EV4B). Although some differences were seen between FOs from lines with similar genotypes, the comparable levels of cortical layers' marker expression between the healthy iPSC lines (WPM F2 and WPM M3) and the disease-bearing lines (WPM D1 and WPM S1) supported the notion of normal neuronal and cortical development (Fig EV4C). Interestingly, RNA levels of Wnt genes did show a pattern suggestive of the Wnt pathway activation (Fig EV4D), which raises a question regarding its role in the pathogenesis of the milder disease. Furthermore, immunostaining and qPCR analyses for astrocytic levels did not reveal significant differences (Fig EV5A and B). Lastly, upon analyzing the DDR signaling in the FOs' VZ, we did not observe major differences in accumulation of DNA damage foci between healthy and sick SCAR12 individuals (Fig EV5C). Altogether, these data suggest different developmental outcomes between SCAR12 and WOREE syndrome-derived organoids.

# Discussion

DEEs are a group of severe neurological syndromes whose underlying molecular pathology is unknown (Howell *et al*, 2021). Together with the lack of accessibility of human samples, it is not surprising that the current medical treatment is lacking. Our study set out to utilize the major technological advances in developmental biology, together with the role of WWOX in the severe WOREE syndrome, to model human refractory DEEs in a tissue-relevant context. By utilizing genetic manipulations and reprogramming, along with electrophysiology, we observed hyperexcitability in both *WWOX* CRISPR-edited and patient-derived brain organoids, therefore successfully demonstrating epileptiform activity. We then further examined the cellular and molecular changes highlighting possible mechanisms for the disease pathophysiology. First, although the neuronal population was largely intact in terms of quantity, we noticed a marked increase in GABAergic markers. This finding is even more surprising when considering the decrease in GABA receptor components seen by RNA-seq. This can indicate a disruption in development of normal and balanced neuronal networks, supporting the increased electrical activity observed in these organoids. It should be noted that several lines of evidence implicate that during development, GABAergic synapses have a depolarizing effect (Obata *et al*, 1978; Ben-Ari *et al*, 2007; Murata & Colonnese, 2020). Seizure dynamics

in developmental epilepsies are known to be dependent on depolarizing GABA responses, particularly due to an accumulation of intracellular chloride resulting in a depolarized chloride reversal potential, thereby causing increased excitability, instead of hyperpolarization upon activation of GABA$_A$ receptors (Khalilov *et al*, 2005; Ben-Ari *et al*, 2007). The evidence of increased mean spectral power in WWOX-depleted COs and WSM FOs, and its recovery in the presence of lentivirus containing WWOX, further strengthens the idea that depolarizing GABA plays a key role in seizure susceptibility. These findings shed a new light on the lack of efficacy of common anticonvulsant therapies on immature neurons (Khalilov *et al*, 2005; Murata & Colonnese, 2020)—making WWOX-depleted COs a useful model to test and study novel therapies targeting excitatory GABAergic responses. An increase in SWO, as demonstrated in Fig 2B, has previously been linked to various stages of the seizure cycle—onset, throughout the seizure, and termination (Bragin & Engel, 2008). Previous studies have also demonstrated that SWOs modulate cortical excitability (Vanhatalo *et al*, 2004) and can localize the seizure-onset zone during the preictal period (Miller *et al*, 2007). Furthermore, slow waves have been identified as characteristic of EEG ictal activity in full-term and preterm infants (Patrizi *et al*, 2003). The mechanism for SWOs during seizure development is poorly understood. However, a few hypotheses suggest an increase in extracellular potassium, changes in the pH, glial cell dysfunction, and/or blood–brain barrier functions (Bragin & Engel, 2008). While exploring the mechanism is beyond the scope of this paper, these are interesting directions to explore in the future.

On the other hand, Fig EV1G shows a decrease in higher frequency activity—beta (12–30 Hz) and gamma (> 30 Hz) oscillations. Although higher frequency activity—particularly gamma oscillations—is known to increase prior to seizure onset at specific focal zones and has been studied as a possible determinant of epileptogenesis (Medvedev *et al*, 2011), our data do not support this for WWOX-related seizures. One possible explanation for this is the maturity of the 7-week-old COs. Gamma oscillations are associated with functional connectivity and integrate neural networks within and across brain structures (Kheiri *et al*, 2013; Ahnaou *et al*, 2017). Lower power in these high-frequency ranges may be due to the delayed development and lower functional connectivity in WWOX-KO organoids. An interesting observation in our week 12 WSM FOs (Appendix Fig S5A and B) was that the activity in WSM S5 FOs was enhanced at high-frequency ranges. This contrast could be due to age, development protocol, or underlying mechanism for detected epileptiform activity. These—particularly the underlying mechanism—would need to be investigated further using in-depth single-cell analysis and through the use of targeted channel blockers and drugs.

Secondly, we closely examined other populations seen in brain organoids and found an increase in astrocytic markers, while the RG population, which express high levels of WWOX, seemed to maintain normal proportions. This pattern was detected early on and appeared to stem from the vRGs, and not from the APCs. A possible explanation is the impaired DDR signaling observed in WWOX-depleted organoids; previous studies in both ESC-derived and primary murine neural stem cells (NSCs) found that accumulation of DNA damage foci, either in the nuclear or in the mitochondrial DNA, causes NSCs to shift to astrocytic differentiation (Wang *et al*, 2011; Schneider *et al*, 2013). In the CNS, physiological DNA breaks can be formed by replicative stress (mainly in dividing progenitor

cells), by oxidative and metabolic stress as a result of accumulation of reactive oxygen species (ROS), and even by neuronal activity (as part of developmental processes and learning) (Suberbielle *et al*, 2013; Madabhushi *et al*, 2014; Madabhushi *et al*, 2015). Impaired repair of these breaks is linked with CNS pathology and neurodegeneration (Suberbielle *et al*, 2013; Madabhushi *et al*, 2014; Shanbhag *et al*, 2019). Our findings suggest a homeostatic role for WWOX in the vRGs, in which WWOX maintains proper DDR signaling in physiological conditions and prevents accumulation of DNA damage associated with impaired differentiation. It is important to note that although we chose to focus on vRGs for the DDR analysis, as they highly express WWOX, our data do not suggest these breaks accumulate specifically in vRGs and could possibly persist in the progenies.

Although the ability of brain organoids to develop functional synapses and complex neural network dynamics is rapidly being established through intensive research (Trujillo *et al*, 2019; Sidhaye & Knoblich, 2020), the capability to model epileptiform activity is only recently being studied (preprint: Samarasinghe *et al*, 2019; Sun *et al*, 2019). Sun *et al* (2019) utilized brain organoids to model Angelman syndrome using UBE3A-KO hESCs, recapitulating hyperactive neuronal firing, aberrant network synchronization, and the underlying channelopathy, which was observed in 2D and mouse models (Sun *et al*, 2019). Samarasinghe *et al* (2019) took advantage of the organoid fusion method and generated organoids enriched with inhibitory interneurons from Rett syndrome patient's iPSCs. In the disease-bearing organoids, they observed susceptibility to hyperexcitability, reductions in the microcircuit clusters, recurring epileptiform spikes, and altered frequency oscillations, which were traced back to dysfunctional inhibitory neurons (preprint: Samarasinghe *et al*, 2019). Furthermore, the model was used to test treatment options by treating the mutated organoids with valproic acid (VPA) or with the TP53 inhibitor, pifithrin-α (PFT), showing improved neuronal activity compared with the treatment with vehicle, with better results using PFT rather than VPA. Although pioneering, these studies focused on the electrophysiological changes seen in the disease-modeling organoids. Considering the lack of gross neurohistological changes in epileptic patients to direct the mechanistic research (Blumcke *et al*, 2017), our study sought to strengthen the utilization of brain organoids for the molecular study of epilepsy. This end was highlighted by bulk RNA-seq analysis, showing defective regional identity acquisition, cortical layer disruption, and Wnt signaling activation. The latter is of particular interest in light of the purposed role for Wnt signaling pathway as a regulator of seizure-induced brain consequences, and therefore a possible target for treatment (Yang *et al*, 2016; Qu *et al*, 2017; Hodges & Lugo, 2018). The forementioned cortical dyslamination is reminiscent of cortical dysplasia, which have a well-recognized role in the pathogenesis of drug-resistant epilepsy (Tassi *et al*, 2002; Fauser *et al*, 2006; Kobow *et al*, 2019).

In agreement with our findings, a recent study that examined the brain histology of a fetus suffering from the WOREE syndrome reported anomalous migration of the external granular layer within the molecular layer of the cortex, a phenotype that was validated also in a rat model with spontaneous WWOX mutations (Iacomino *et al*, 2020). This observation is further supported by the transcriptomic analysis performed by Kosla *et al* (2019) on human neuronal progenitor cells (hNPCs) after silencing *WWOX* using shRNA. This study found that knocking down WWOX causes hNPCs to lose the enrichment of genes related to neural crest differentiation and migration and to cell–cell adhesion, present in WT hNPCs. The authors also reported decreased mitochondrial redox potential, enhanced cellular adhesion to the growth surface, and reduced expression of MMP2 and MMP9. Iacomino *et al* (2020) reanalyzed this transcriptomic data, focusing on genes associated with neuronal migration and differentiation, and found reduced expression of some neural migration-related genes, such as microtubule proteins and kinesin family proteins. Notably, cortical layering was found to be affected by the status of the Wnt pathway (Qian *et al*, 2020), a pathway in which WWOX has been implicated through its binding partners. For example, WWOX was found to bind the Dishevelled proteins Dvl1 and Dvl2, with the latter being inhibited by WWOX, therefore attenuating the Wnt pathway (Bouteille *et al*, 2009; Abu-Odeh *et al*, 2014a). Our study further highlights a possible crosstalk between Wnt activation and DNA damage, a phenomenon that was previously described (Elyada *et al*, 2011). This is very much in line with the previously described pleiotropic functions of WWOX (Abu-Remaileh *et al*, 2015) and with the reduced negative regulation of cell cycle and MDM2 levels seen in our RNA-seq. We found accumulation of DNA breaks in Ki67$^+$ cells in the VZ of KO COs, which might be explained by Wnt activation, promoting proliferation and likely replicative stress.

In addition to disease modeling in brain organoids, we attempted to rescue the phenotypes seen by reintroducing WWOX to the hESCs genome. This resulted in supraphysiological expression of WWOX in all cell populations seen in COs and a partial rescue. These results provide a proof of concept for successful reintroduction of WWOX as a mean for correcting the phenotype and possibly for therapeutic intervention. Yet, our findings suggest the importance of optimizing population-targeted delivery and fine-tuning of expression levels for successful genetic therapy approaches in patients with WOREE syndrome.

Lastly, we generated FOs from patients suffering from WOREE syndrome (WSM) and the relatively milder phenotype—SCAR12 (WPM). Our findings indicate that both brain organoid culture protocols (COs and FOs) result in similar outcomes, validating the phenotype of WWOX deficiency and that it stems from cortex. Intriguingly, when modeling the family of SCAR12, we did not observe the same developmental abnormalities as in WOREE organoids. SCAR12 FOs exhibited very mild, if any, differences in the forebrain neuronal population development, astrocyte development, and DDR signaling. This strengthens the system's ability to model the differences seen between the syndromes, and points out the need of closer examination of the rare SCAR12 syndrome and the pleiotropic functions of WWOX (Abu-Remaileh *et al*, 2015; Banne *et al*, 2021). It is noteworthy that although there is a marked difference in WWOX expression in the healthy heterozygote parents from different families, there is a very minor difference in the levels observed in the affected homozygote patients (Figs 5A and EV4A, Appendix Figs S4A and S6A). These results raise the question whether the disease severity is correlated with the functional levels of WWOX rather than the total expression levels.

Overall, our data demonstrate the ability of brain organoids to model childhood epileptic encephalopathies, while elucidating the pathological changes seen in patients with germline mutations of *WWOX* and possible approaches for treatment development.

### Limitations of study

As samples from DEE patients in general, and WWOX-related disorders in particular, are limited, we generated patient-derived organoids from only two families, one of each syndrome. This makes generalizing our results more difficult, a problem we partially addressed by gene manipulation in hESCs. Another limitation stems from the well-described heterogeneity of brain organoids, which we dealt with by analyzing several repeats and confirmed in patient-derived models and protocols.

# Materials and Methods

### Cell culture and plasmids

WiBR3 hES cell line and the generated iPS cell lines were maintained in 5% $CO_2$ conditions on irradiated DR4 mouse embryonic fibroblast (MEF) feeder layers in FGF/KOSR conditions: DMEM/F12 (Gibco; 21331-020 or Biological Industries; 01-170-1A) supplemented with 15% knockout serum replacement (KOSR, Gibco; 10828-028), 1% GlutaMAX (Gibco; 35050-038), 1% MEM non-essential amino acids (NEAA, Biological Industries; 01-340-1B), 1% sodium pyruvate (Biological Industries; 03-042-1B), 1% penicillin–streptomycin (Biological Industries; 03-031-113), and 8 ng/ml bFGF (PeproTech; 100-18B). Medium was changed daily, and cultures were passaged every 5–7 days either manually or by trypsinization with trypsin type C (Biological Industries; 03-053-1B). Rho-associated kinase inhibitor (ROCKi, also known as Y27632) (Cayman; 10005583) was added for the first 24–48 h after passaging at a 10 μM concentration.

For transfection of hESCs, cells were cultured in 10 μM ROCKi 24h before electroporation. Cells were detached using trypsin C solution and resuspended in PBS (with $Ca^{2+}$ and $Mg^{2+}$) mixed with a total of 100 μg DNA constructs, and electroporated in Gene Pulser Xcell System (Bio-Rad; 250 V, 500 μF, 0.4-cm cuvettes). Cells were subsequently plated on MEF feeder layers in FGF/KOSR medium supplemented with ROCKi. For WWOX-KO, px330 plasmid containing the sgRNA targeting exon 1 was co-electroporated in 1:5 ratio with pNTK-GFP, and 48hr later, GFP-positive cells were sorted and subsequently plated sparsely (2,000 cells per 10-cm plate) on MEF feeder plates for colony isolation, ~10 days later. For WWOX reintroduction, pAAVS-2aNeo-UBp-IRES-GFP plasmid cloned to carry the WWOX coding sequence was co-electroporated with px330 targeting the AAVS1 locus (Guernet et al, 2016), sorted for GFP, and selected with 0.5 μg/ml puromycin for colony isolation. Gene editing was validated via Western blot. sgRNA sequences are noted in Table EV3.

For RNA or protein isolation, hPSCs were passaged onto Matrigel-coated plates (Corning; 356231) as indicated above and were cultured in NutriStem hPSC XF Medium (Biological Industries; 05-100-1A).

### Cerebral organoid generation, culture, and lentiviral infection

Cerebral organoids were generated from hESCs as previously described (Lancaster et al, 2013; Lancaster & Knoblich, 2014; Bagley et al, 2017; Lancaster et al, 2018), with the following changes:

Human WiBR3 cells and WSM iPSCs were maintained on mitotically inactivated MEFs. 4–7 days before protocol initiation, cells were passaged onto 60-mm plates coated with either MEFs or Matrigel (Corning; FAL356231) and grown until 70–80% confluency was reached. On day 0, hESC colonies were detached from MEFs with 0.7 mg/ml collagenase D solution (Sigma; 11088858001) and dissociated to single-cell suspension using a quick 2-min treatment with trypsin type C. For cells cultured on Matrigel, collagenase D treatment was skipped, and cells were immediately dissociated with trypsin type C, with no other variations in protocols from this point forward. Although only empirically observed, no major differences were seen in final outcome; however, MEF-cultured hPSCs seemed to have better success rates of neural induction and therefore were preferentially used.

After dissociation, cells were counted and suspended in hESC medium, composed of DMEM/F12-supplemented 20% KOSR, 3% USDA-certified hESC-quality FBS (Biological Industries), 1% GlutaMAX, 1% NEAA, 100 μM 2-mercaptoethanol (Sigma; M3148), 4 ng/ml bFGF, and 10 μM Rocki. For embryoid body (EB) formation 9,000 cells were seeded in each well of an ultra-low attachment V-bottom 96-well plates (S-Bio Prime; MS-9096VZ). EBs were fed every other day for another 5 days, in which fresh bFGF and ROCKi were added in the first change. At day 6, the medium was replaced with Neural Induction (NI) medium (Bagley et al, 2017), composed of DMEM/F12, 1% N2 supplement (Gibco; 17502048), 1% GlutaMAX, 1% MEM-NEAA, and 1 μg/ml heparin solution (Sigma; H3149). NI medium was changed every other day until establishment of neuroepithelium (usually on days 11–12), where quality control was performed as indicated (Lancaster & Knoblich, 2014; Bagley et al, 2017), and well-developed EBs were embedded in Matrigel droplets (Lancaster & Knoblich, 2014; Bagley et al, 2017). Droplets were transferred to 90-mm sterile, non-treated, culture dishes (Miniplast; 825-090-15-017) with Cerebral Differentiation Medium (CDM) composed of 1:1 mixture of DMEM/F12 and Neuro-basal Medium (Gibco; 21103049 or Biological Industries; 06-1055-110-1A), 0.5% N2 supplement, 1% B27 supplement without vitamin A (Gibco; 12587010), 1% GlutaMax, 1% penicillin/streptomycin, 0.5% NEAA, 50 μM 2-mercaptoethanol, 2.5 μg/ml human recombinant Insulin (Biological Industries; 41-975-100), and 3 μM CHIR-99021 (Axon Medchem; 1386). The medium was changed every other day. From day 16 onward, organoids were cultured on an orbital shaker at 37°C and 5% $CO_2$ in Cerebral Maturation Medium (CMM) (Lancaster et al, 2018) composed similar to CDM, with B27 supplement changed to B27 supplement containing vitamin A (Gibco; 17504044), without CHIR-99021, and containing 400 μM vitamin C (Sigma; A4403) and 12.5 mM HEPES buffer (Biological Industries; 03-025-1B). Medium was changed every 2–4 days. From week 6, 1% Matrigel was added to the medium. To reduce chances of contamination, every 30 days the organoids were moved to fresh sterile plates. All of the described media were filtered through a 0.22-μm filter and stored at 4°C until usage. For all analyses, organoids from the same batch were used, unless stated otherwise.

Lentiviral transduction of WWOX was carried as previously published (Deverman et al, 2016; Khawaled et al, 2019). Briefly, viruses carrying WWOX were generated from pDEST12.2TM destination vector (Gateway Cloning Technology). After ultracentrifugation, titer was determined empirically by infecting 293T cells. At day 35 of culture, individual COs were transferred to an Eppendorf

tube containing CMM with 1:100 of virus-containing medium and 5 μg/ml polybrene (Merck; TR-1003-6) and incubated overnight. The day after, organoids were put back on shaking culture with fresh medium.

### Reprogramming of somatic cells

Blood samples from families affected by WOREE and SCAR12 syndromes were donated under the approval of the Kaplan Medical Center Helsinki Committee for research purposes only, with informed consent obtained from all human subjects, and all the experiments conformed to the principles set out in the WMA Declaration of Helsinki and the Department of Health and Human Services Belmont Report.

Derivation of iPSCs directly from PBMCs was conducted by infection with the Yamanaka factors and Sendai virus CytoTune-iPS 2.0 Kit according to the manufacturer's instructions. Briefly, blood samples from PBMCs were isolated by Ficoll gradient and were cultured with StemPro-34™ medium (Gibco; 10639-011) supplemented with StemPro-34 Nutrient Supplement (Gibco; 10639-011), 100 ng/ml human SCF (PeproTech; 300-07), 100 ng/ml human FLT-3 ligand (R&D Systems; 308-FKE), 20 ng/ml human IL-3 (PeproTech; 200-03), and 10 ng/ml Human IL-6 (PeproTech; 200-06). After 24 h, half of the medium was replaced. After additional 24 h, day 0 of the protocol, cells were transferred to 6-well plates, reprogramming virus mixture was added, and the plates were centrifuged at 1,000×*g* for 30 min at room temperature. Cells were resuspended and placed back in the incubator overnight. The next day, to get rid of the remaining virus, the cells were centrifuged washed and resuspended in fully supplemented StemPro-34 medium, with extra medium addition on day 2. On day 3, cells were transferred to 10 cm MEF-coated plates, with half the medium replaced with complete StemPro-34 without cytokines and half medium changes every other day. By day 7, cells in different phases of reprogramming were seen, and the medium was gradually changed into mTeSR supplemented with 10 μM ROCKi to prevent reprogramming-related apoptosis. On day 16, colonies with normal morphology and growth rate were picked, expanded, validated for expression of pluripotency markers, and sequenced for WWOX mutations.

### Forebrain organoid generation and culture

Forebrain organoids were generated from iPSCs as previously described (Qian *et al*, ,2016, 2018), with the changes noted below:

iPSC cells were maintained on mitotically inactivated MEFs. 4–7 days before protocol initiation, cells were passaged onto MEF-coated 60 mm plates and were cultured up to 70-80% confluency. On day 0, iPSC colonies were detached, dissociated, and counted the same as for COs, and resuspended in hPSC medium containing DMEM/F12, 20% KOSR, 1% GlutaMax, 1% MEM-NEAA, 1% penicillin/streptomycin, and 100 μM 2-mercaptoethanol. 9,000 cells per well were seeded in V-bottom 96-well plate. On day 1, medium was changed to Neuroectoderm Medium (NEM), which is hPSC medium freshly supplemented with 2 μM A83 (Axon Medchem; 1421) and 100 nM LDN-193189 (Axon Medchem; 1527), which was changed every other day. On days 5 and 6, half of the medium was aspirated and replaced by Neural Induction Medium (NIM) composed of

DMEM/F12, 1% N2 supplement, 1% GlutaMax, 1% penicillin/streptomycin, 1% NEAA, 10 μg/ml heparin, 1 μM CHIR-99021 (Axon Medchem; 1386), and 1 μM SB-431542 (Sigma; S4317). On day 7, quality control and Matrigel embedding were performed as indicated (Qian *et al*, 2018), and EBs were continued to be cultured in NIM with medium changes every other day. At day 14, Matrigel removal was preformed (Qian *et al*, 2018), medium was changed to Forebrain Differentiation Medium (FDM) composed of DMEM/F12, 1% N2 supplement, 1% B27 with vitamin A, 1% NEAA, 1% GlutaMax, 1% penicillin/streptomycin, 50 μM 2-mercaptoethanol, and 2.5 μg/ml insulin, and transferred to an orbital shaker at 37°C and 5% $CO_2$. Medium was changed every 2–3 days. On day 71, the medium was changed to Forebrain Maturation Medium (FMM), containing Neurobasal medium, 1% B27 supplement with vitamin A, 1% GlutaMax, 1% penicillin/streptomycin, 50 μM 2-mercaptoethanol, 200 μM vitamin C, 20 ng/ml human recombinant BDNF (PeproTech; 450-02), 20 ng/ml human recombinant GDNF (PeproTech; 450-10), 1 μM dibutyryl-cAMP (Sigma; D0627), and 1 ng/mL TGF-β1 (PeproTech; 100-21C). Medium was changed every 2–3 days.

### Immunofluorescence

Organoid fixation and immunostaining were performed as previously described (Mansour *et al*, 2018). Briefly, organoids were washed three times in PBS, then transferred for fixation in 4% ice-cold paraformaldehyde for 45 min, washed three times in cold PBS, and cryoprotected by overnight equilibration in 30% sucrose solution. The next day, organoids were embedded in OCT, snap-frozen on dry ice, and sectioned at 10 μm by Leica CM1950 cryostats.

For immunofluorescent staining, sections were warmed to room temperature and washed in PBS for rehydration, permeabilized in 0.1% Triton X-100 in PBS (PBT), and then blocked for 1 hr in a blocking buffer containing 5% normal goat serum (NGS) and 0.5% BSA in PBT. The sections were then incubated at 4°C overnight with primary antibodies diluted in the blocking solution. The day after, sections were then washed in three times while shaking in PBS containing 0.05% Tween-20 (PBST) and incubated with secondary antibodies and Hoechst 33258 solution diluted in blocking buffer for 1.5 h at RT. Slides were washed four times in PBST while shaking, and coverslips were mounted using Immunofluorescence Mounting Medium (Dako; s3023). Sections were imaged with Olympus FLUO-VIEW FV1000 confocal laser scanning microscope and processed using the associated Olympus FLUOVIEW software. γH2AX-positive nuclei were manually counted using NIH ImageJ and statistically analyzed as later described.

A list of primary and secondary antibodies used in this work, together with dilutions details, can be found in Table EV4.

### Electrophysiological recordings

Organoids were embedded in 3% low-temperature gelling agarose (at ~36°C) and incubated on ice for 5 min, after which they were sliced to 400 μm using a Leica 1200S Vibratome in sucrose solution (in mM: 87 NaCl, 25 $NaHCO_3$, 2.5 KCl, 25 glucose, 0.5 $CaCl_2$, 7 $MgCl_2$, 1.25 $NaHPO_4$, and 75 sucrose) at 4°C. Slices were incubated in artificial cerebrospinal fluid (ACSF, in mM: 125 NaCl, 25 $NaHCO_3$, 2.5 KCl, 10 glucose, 2.5 $CaCl_2$, 1.5 $MgCl_2$, pH 7.38, and 300 mOsm) for 30 min at 37°C, followed by 1 h at RT. During

recordings, slices were incubated in the same ACSF at 37°C with perfused carbogen (95% $O_2$, 5% $CO_2$), in baseline condition. Local field potential (LFP) and whole-cell patch-clamp recordings were done using electrodes pulled from borosilicate capillary glass and positioned 150-μm deep from the outer rim of each slice (see Fig EV1F). LFP electrodes were filled with ACSF, while patch electrodes were filled with internal solution. Data were recorded using MultiClamp software at a sampling rate of 25,000 Hz. Data were analyzed using MATLAB software. Traces were filtered using (i) 60 notch filter (with 5 harmonics) to eliminate noise and (ii) 0.1-Hz high-pass IIR filter to eliminate fluctuations from the recording setup. The detrended feature (using the hamming window) was then used to eliminate large variations in the signal, and the normalized spectral power was calculated using the fast Fourier transform. The area under the curve of the power spectral density plots was calculated by taking the sum of binned frequencies over specific frequency ranges.

### Cell-attached recordings

Cell-attached recordings were obtained with blind patch-clamp recordings. We recorded spontaneous neuronal activity from organoids' neuronal populations. Electrodes (~7 MOhm) were pulled from filamented, thin-walled, borosilicate glass (outer diameter, 1.5 mm; inner diameter, 0.86 mm; Hilgenberg GmbH) on a vertical two-stage puller (PC-12, Narishige). The electrodes were filled with an internal solution that contained the following (in mM): 140 K-gluconate, 10 KCl, 10 HEPES, 10 $Na_2$-phosphocreatine, and 0.5 EGTA, and adjusted to pH 7.25 with KOH.

The electrodes were inserted at $45^0$ to the organoid's surface. During the recordings, the organoids were kept in CMM without Matrigel at 35°C. An increase in the pipette resistance to 10–200 MOhm resulted in most cases in the appearance of spikes. The detection of a single spike was the criteria to start the recording. All recordings were acquired with an intracellular amplifier in current-clamp mode (MultiClamp 700B, Molecular Devices), acquired at a sampling rate of 10 kHz (CED Micro 1401-3, Cambridge Electronic Design Limited), and filtered with a high-pass filter to eliminate field potentials and retain neuronal spikes.

Data analysis of the cell-attached recordings was carried out with custom-written code in MATLAB (The MathWorks). Spikes recorded in the cell-attached mode were extracted from raw voltage traces by applying a threshold (the spikes' threshold was placed well above the peaks in the background noise level). For calculating the average firing rate, the firing rate over a 4-min recording period was calculated for each recorded cell.

### Immunoblot analysis and subcellular fractionation

For total protein, organoids were homogenized in lysis buffer containing 50 mM Tris (pH 7.5), 150 mM NaCl, 10% glycerol, and 0.5% Nonidet P-40 (NP-40) that was supplemented with protease and phosphatase inhibitors. For separation of cytoplasmic fraction, organoids were grinded in a hypotonic lysis buffer [10 mmol/l HEPES (pH 7.9), 10 mmol/l KCl, 0.1 mmol/l EDTA] supplemented with 1 mmol/l DTT and protease and phosphatase inhibitors. The cells were allowed to swell on ice for 15 min, then 0.5% NP-40 was added, and cells were lysed by vortex. After centrifugation, the cytoplasmic

fraction was collected. Afterwards, nuclear fraction was obtained by incubating remaining pellet in a hypertonic nuclear extraction buffer [20 mmol/l HEPES (pH 7.9), 0.42 mol/l KCl, 1 mmol/l EDTA] supplemented with 1 mmol/l DTT for 15 min at 4°C while shaking. The samples were centrifuged, and liquid phase was collected.

Western blotting was performed under standard conditions, with 40–50 μg protein used for each sample. Blots were repeated and quantified 2–3 times per experiment in Bio-Rad's Image Lab software. Representative images of those repeated experiments are shown.

### RNA extraction, reverse transcription–PCR, and qPCR

Total RNA was isolated using Bio-Tri reagent (Biolab; 9010233100) as described by the manufacturer for phenol/chloroform-based method. 0.5–1 μg of RNA was used to synthesize cDNA using a qScript cDNA Synthesis Kit (QuantaBio; 95047). qRT–PCR was performed using Power SYBR Green PCR Master Mix (Applied Biosystems; AB4367659). All measurements were performed in triplicate and were standardized to the levels of either HPRT or UBC. All primer sequences used are noted in Table EV5.

### Library preparation and RNA sequencing

Library preparation and RNA sequencing was performed by the Genomic Applications Laboratory in the Hebrew University's Core Research Facility following the standard procedures. Briefly, RNA quality was assessed by using RNA ScreenTape Kit (Agilent Technologies; 5067-5576), D1000 ScreenTape Kit (Agilent Technologies; 5067-5582), Qubit(r) RNA HS Assay Kit (Invitrogen; Q32852), and Qubit(r) DNA HS Assay Kit (Invitrogen; 32854).

For mRNA library preparation, 1 μg of RNA per sample was processed using KAPA Stranded mRNA-Seq Kit with mRNA Capture Beads (Kapa Biosystems; KK8421). Library was eluted in 20 μl of elution buffer and adjusted to 10 mM, and then, 10 μl (50%) from each sample was collected and pooled in one tube. Multiplex sample pool (1.5pM including PhiX 1.5%) was loaded in NextSeq 500/550 High Output v2 Kit (75 cycles) cartridge (Illumina; FC-404-1005) and loaded on NextSeq 500 System Machine (Illumina), with 75 cycles and single-read sequencing conditions.

For library quality control, Fastq files were tested with *FastQC* (ver.0.11.8) and trimmed for residual adapters, low-quality bases (Q = 20), and read length (20 bases). Trimming was performed with *trim galore* (ver.0.6.1). Read counts were high around 30–50 M per sample and decreased negligibly after filtering. Transcriptome mapping was performed with *salmon* (ver.1.2.1) in its mapping-based mode, turning on both validate mapping mode and gc-bias correction. Prior to alignment, a salmon index was created based on HS GRCh38 CDNA release 99 (Nov 2019) using kmer size of 25. Salmon mapping reports both raw transcripts count and TPM counts. Resulting mapping rates are high between 80% and 90%. A total of 8 CO samples were sequenced (4 WT COs and 4 KO COs)—one WT sample failed our preliminary quality control (low read count and low transcriptome mapping rate). Another WT sample that did not cluster with any of the other samples (neither WWOX-KO nor WT) was apparent in both PCA and dendrogram analysis. These two samples were extracted from further analysis, giving a total of six samples used for further analysis. For differentially

expressed gene determination (KO versus WT), raw transcript counts were filtered for minimal overall count of 10 on all six samples and imported with R package *tximport* (ver.1.16.1) for analysis with *DEeq2* (ver.1.28.1). Counts were normalized by DESeq2, and differentially expressed genes were filtered, setting alpha to 0.01. Mean-based fold change was calculated, as well as a shrink-based fold change, based on *apeglm* (ver.1.10.0). The resulting set of 15,370 genes is illustrated in a "volcano scatter plot" showing fold change against *P*-values (Fig EV3B).

For the preperation of the heatmaps shown in Fig 4A and C, the list of differentially expressed genes was separated to upregulated (WWOX-KO expression was higher than WT expression) and downregulated sublists. Each sublist was sorted by fold change values, and top 100 genes were selected from each sublist. For each of the selected genes, log2-normalized counts were scaled and presented in a heatmap using heatmap.2 from R package *gplots* (ver.3.0.3).

For the heatmap seen in Fig 4F, log2-normalized counts for each of the six cortical layer gene markers were scaled and presented in a heatmap form using heatmap.2 from R package *gplots*. For enrichment plot (Fig 4B and Appendix Fig S3B), gene set enrichment analysis was performed with Broad Institute *GSEA* software (ver.4.0.3). Input included 15,348 genes ranked by log2 of fold change. GO sets are Broad Institute set c5.all.v7.0. Permissible sets are those with at least 15 genes and no more than 500 genes. For enrichment plot seen in Appendix Fig S3A, gene set enrichment analysis was performed with *WebGestalt*R (ver.0.4.3). Input includes 3,000 differentially expressed genes with most significant adjusted *P*-values (*P*-value threshold of 0.02) and ranked by log2 of fold change. Gene sets are GO biological processes. Permissible sets in this analysis are those with at least 10 genes and no more than 500 genes. PCA plot (Fig EV3) of first two components was calculated and plotted with base R functions. Calculation is based on log2-transformed and log2-normalized counts adding pseudo count of 1.

## Statistics

Results of the experiments were expressed either as mean ± SEM or in a boxplot indicating the 1$^{st}$ and 3$^{rd}$ quartiles, minimum and maximum values, and the median. First, the Wilk–Shapiro test was used to determine normality: For normally distributed samples, a two-tailed unpaired Student's *t*-test with Welch's correction was used to compare the values of the test and control samples. For non-normally distributed samples, the non-parametric Mann–Whitney test was used. For comparisons between more than two samples, one-way ANOVA was used, correcting for the multiple comparisons with Tukey's multiple comparisons test. For samples that were not normally distributed, the Kruskal–Wallis test was used with Dunn's multiple comparisons test. For the kinetic experiments (Fig EV3C), the analysis was corrected for multiple *t*-tests using the Holm–Šídák method, without assuming equal SD. *P*-value cutoff for statistically significant results was as follows: n.s (non-significant), *$P \leq 0.05$, **$P \leq 0.01$, ***$P \leq 0.001$, and ****$P \leq 0.0001$. Statistical analysis and visual data presentation were preformed using GraphPad Prism 8. Exact *P*-values and the specific tests used are stated in Appendix Table S1. No randomization or blinding was applied in this study. The experiments were performed on several biological replicates, with at

### The paper explained

**Problem**
Developmental and epileptic encephalopathy (DEE) is a group of disorders associated with intractable seizures, brain development, and functional abnormalities, and, in some cases, premature death. Germline mutations in tumor suppressor gene *WWOX* cause severe neurological syndromes termed *WWOX*-related epileptic encephalopathies (WOREE syndrome) and spinocerebellar ataxia type 12 (SCAR12 syndrome). Very little is known about the role of *WWOX* in the brain and the molecular consequences of *WWOX* mutations, rendering the patients poorly treated and severely impaired.

**Results**
In this study, we generated an in vitro model for studying DEEs. Through the application of gene editing and reprogramming of patient-derived samples, we generated human pluripotent stem cells harboring *WWOX* mutations. These were subsequently used to generate brain organoids that model the neurological pathology. This enabled us to discover previously unknown human molecular defects, such as imbalance between glutamatergic and GABAergic neurons, enhanced astrogenesis, defects in the DNA damage response, cortical dysplasia, and chronic activation of Wnt. Lastly, by reintroducing *WWOX* we could potentially rescue these phenotypes.

**Impact**
As time passes since the first description of *WWOX*-related neurological DEE in 2014, the number of patients diagnosed with *WWOX*-related disorders is rising rapidly. These children suffer from severe symptoms and in most cases premature death. This study opens a new avenue for studying DEEs in a human context, an insight that was previously rare. Moreover, it serves as a platform to study possible therapeutic intervention, and although further study is required before clinical translation, our data serve as a proof of concept for the use of genetic therapy in the setting of *WWOX* mutations.

least two hPSC lines used for each genotype (with the exception of the WiBR3 WT line). Unless stated otherwise, the experiments were performed on multiple batches of organoids.

## Data and code availability

The RNA-seq dataset produced in this study is available in the following databases:

RNA-seq data for week 15 cerebral organoids: Gene Expression Omnibus: GSE156243 (https://www.ncbi.nlm.nih.gov/geo/query/acc.cgi?acc=GSE156243).

*Expanded View* for this article is available online.

## Acknowledgements

We would like to thank all members of the Aqeilan's laboratory for fruitful discussion and Jonathan Monin for his invaluable bioinformatic analysis. We are grateful to Dr. Abed Nasereddin and Dr. Idit Shiff from the Genomic Core Facility for their help. We would also like to thank Prof. Eli Pikarsky from the Hebrew University for his guidance in the histological analysis, and Dr. Jonathan Bayerl and Dr. Venkat Raghavan Krishnaswamy from Weizmann Institute of Science for their support in stem cells and organoid cultures. The Aqeilan's laboratory is funded by the European Research Council (ERC) [No.

682118] and Proof-of-Concept ERC Grant [No. 957543]. Shani Stern is supported by the Zuckerman STEM Leadership Program.

## Author contributions

DJS, SR, and RIA conceptualized the study. DJS, SR, JHH, and RIA contributed to methodology. DJS, IK, BA, SV, and AS performed investigation. DJS, AS, SS, PLC, and RIA wrote, reviewed, and edited the manuscript. RIA performed funding acquisition. EB, MM, and JHH provided resources. RIA contributed to project administration. SS, PLC, and RIA underwent supervision.

## Conflict of interest

The authors declare that they have no conflict of interest.

## For more information

i OMIM website of DEE28 at https://www.omim.org/entry/616211

ii Human disease genes website at https://humandiseasegenes.nl/wwox/

iii Visit the "WWOX Foundation" official website at https://www.wwox.org/

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
