## [Review Process File · EMBO Molecular Medicine]

Modeling Genetic Epileptic Encephalopathies using Brain Organoids

Daniel Steinberg, Srinivasa Repudi, Afifa Saleem, Irina Kustanovich, Sergey Viukov, Baraa Abudiab, Ehud Banne, Muhammad Mahajnah, Jacob Hanna, Shani Stern, Peter Carlen, and Rami Aqeilan
DOI: [10.15252/emmm.202013610](https://doi.org/10.15252/emmm.202013610)

Corresponding author: Rami Aqeilan (ramiaq@mail.huji.ac.il)

Review Timeline:	Transfer from Review Commons:	19th Oct 20
	Editorial Decision:	22nd Oct 20
	Revision Received:	21st Apr 21
	Editorial Decision:	11th May 21
	Revision Received:	3rd Jun 21
	Accepted:	4th Jun 21

Editor: Zeljko Durdevic

Review
COMMONS

Transaction Report: This manuscript was transferred to EMBO Molecular Medicine following peer review at Review Commons

22nd Oct 2020

Dear Prof. Aqeilan,

Thank you for the submission of your manuscript to EMBO Molecular Medicine. I have now had a chance to carefully read your point-by-point response. I also discussed your work and your response to the referees' comments with other members of our editorial team. As all three reviewers are generally supportive of your study, together with your well-thought-out revision plan we would like to invite major revision of your manuscript. Please also include following points suggested by the referees in your revision :

- Referee #1:

- o Please confirm main findings in patient-derived cortical organoids (similar to point 4 of referee #3).
- o Co-staining with beta-catenin and Nestin is welcomed but not required.
- o Discuss microcephaly in WOREE context. Organoids' diameter measurement is welcomed but not required (similar to point 4 of referee #2).

- Referee #3:

- o scRNA sequencing experiments are NOT required. Please perform detailed/extended analyses using qPCR, immunostaining, FACS etc. to, for example, quantify levels of different cells in organoids or to address cortical layering.
- o Please perform a new patient-derived cortical organoid batch (similar to point 9 of referee #1).

Addressing the reviewers' concerns in full, in writing or experimentally, will be necessary for further considering the manuscript in our journal, and acceptance of the manuscript will entail a second round of review. EMBO Molecular Medicine encourages a single round of revision only and therefore, acceptance or rejection of the manuscript will depend on the completeness of your responses included in the next, final version of the manuscript. For this reason, and to save you from any frustrations in the end, I would strongly advise against returning an incomplete revision.

We would welcome the submission of a revised version within three to six months for further consideration. However, we realize that the current situation is exceptional on the account of the COVID-19/SARS-CoV-2 pandemic. Please let us know if you require longer to complete the revision.

I look forward to receiving your revised manuscript.

Yours sincerely,

Zeljko Durdevic

Review #1

1. How much time do you estimate the authors will need to complete the suggested revisions:

Estimated time to Complete Revisions (Required)

(Decision Recommendation)

Between 3 and 6 months

2. Evidence, reproducibility and clarity:

Evidence, reproducibility and clarity (Required)

In this manuscript Steinberg et al model with brain organoids the early infantile WWOX-related epileptic encephalopathy and investigate the role of WWOX in embryonic human brain development. They claim that deletion of WWOX gene has no major defects in the different neural progenitor cells but affects specifically interneurons and glial cells. They also claim that these changes are causative for the epileptiform activity observed in the KO COs while mechanistically they show that DNA damage response is affected. Interestingly, they can reverse/rescue the phenotype by re-introduction of the WWOX gene in COs. By performing RNA-seq they show that the Wnt pathway is involved in the pathophysiology of the disease. Finally they claim that patient derived organoids exhibit similar phenotype with the KO COs in terms of cellular composition, neuronal activity and DNA damage response and they suggest that these cellular and molecular differences could be important for the differences in the severity of the WOREE and SCAR12 syndromes which are both developed after mutations in the WWOX gene. Although, the authors present the data clearly and consistently, in my opinion this manuscript would benefit from a more rigorous and deep data analysis. Major concerns also are raised by the relevance of the description on the one hand of the role of WWOX in human brain development and the phenotype described in KO COs and on the other hand of the WOREE and SCAR12 syndromes and the modelling of the diseases. These concerns and further suggestions for improving this work are listed below.

****Major comments:****

1. Regarding the expression of WWOX in the apical side of the neuroepithelium, is it located in the apical belt or it is rather in the cytoplasm of the VZ progenitors? Co-staining with the nuclear marker SOX2 is not helpful to distinguish this. Co-staining with F-actin, b-catenin or other apical belt markers from the one hand and with Nestin from the other would be useful.
2. The authors mention that no changed in the total number of SOX2, TBR2 or NEUN were observed. However, considering the big variability between the number and size of ventricles in COs it is not clear how the authors draw these conclusions. Quantifications of the size of the VZ/CP and absolute numbers of the markers shown in Fig.1C would be useful. This will back up the RNA levels that the authors show in Fig.1F which however are inconclusive regarding the actual size of the ventricles within the COs.
3. The authors show that in the KO COs there is big increase of CAD1 and CAD2 genes.

Taking into account that these experiments were done in unpattern COs, is this correlated with general change in the identity of the cells to ventral? Are also ventral progenitor markers affected (e.g. Nkx2.1)? This could be potentially addressed by analysis of the RNA-seq data that the authors show in Fig.4

4. The authors mention in the introduction that WOREE-patients may have microcephaly. It would be interesting if the authors check whether there are any changes in the total size of the COs either in the KO or in the patient-derived FOs.

5. The rescue of the phenotype observed from the W-AAV COs is interesting. However, more detailed analysis needs to be done comparing the neural markers expression of W-AAV with the ctrl COs. This would be important since the authors propose the use of W-AAV as a rescue strategy. It would be also important to include in this analysis quantifications of the protein levels of the neural markers by stainings or by western blot.

6. It is unclear how/why the authors conclude that the reduction of the proliferating rate of SOX2+ progenitors lead to increase of astrocytes as mentioned in the last sentence of page 9. The increased differentiation of SOX2+ cells may lead to increased neuronal differentiation since authors show also increased neuronal markers in Fig.1. Authors may think of electroporating the KO COs with GFP or to KD the WWOX gene in ctrl organoids via electroporation and follow their progeny. This is an important experiment in my opinion.

7. The finding with increased DDR is very interesting. Is this increase linked with apoptosis as the authors imply? Caspase-3 or tunnel staining would be useful. If so, how the authors explain the similar numbers of SOX2+ progenitors in the KO COs?

8. One major concern regarding the modelling of the WOREE syndrome and the role of WWOX in this syndrome and the human cortical formation is the fact that the patients-derived iPSCs from the sick son (WSMS5) have similar WWOX protein levels with the healthy father and these levels are very much reduced compared to wt (Fig.5A). Should the authors consider using at least the WSMM2 which resembles closely the wt situation? The same situation is observed also in the second family WPM. Fig. S5A doesn't include wt iPSCs while the quality of the pictures of the western blot in the D1 and S1 is bad quality. If the protein is present how can the authors correlate the phenotype observed with the KO COs? If the authors believe that the WWOX in the patients is not functional this needs to be addressed.

9. Another major concern regarding experimental design of the study: It is unclear why the authors decide to generate forebrain organoids when using the patients-derived iPSCs and not continue with the COs since their aim is to compare the patients' phenotype with the KO. And if they want to generate more fate driven organoids why they do not choose the dorsal and ventral organoids published from (Pasca and Knoblich labs) since they do see major differences in the inhibitory vs excitatory neurons (Fig.1). In my opinion the authors should include at least part of the analysis in the unpattern organoids and/or in ventral and dorsal assembloids/fused organoids.

10. Contrast to what authors claim, VGlut1 doesn't seem similar between the affected son and the unaffected father in Fig.5D. In addition, GAD67 staining is not clear from the pictures. Quantifications and a more representative pictures and magnifications are needed.

11. It is not clear why the authors claim that the SCAR12 individuals do not show differences in the astrocytes and DDR. According to Fig.S6I,J,K the WPMS1 shows increased astrocytic markers (from quantifications, GFAP and ALDH1A1) and increased 53BP1 and γ H2AX (staining). This patient also has the lowest WWOX expression according to Fig.S6A. In my view this is the patient who resembles the most the KO situation. Thus, the conclusions made from the authors are contradicting.

Minor comments:

1. Discuss previous publications on the role of WWOX in human brain and human neural progenitor cells. Special emphasis should be given to the comparison of the DEGs in these publications and in the present manuscript.
2. Show WWOX protein levels in the W-AAV COs compared to ctrl levels in Fig S1D.
3. Page 7 in the paragraph: "These findings suggest that during human embryonic....." are the authors mean "apical layer of the VZ" instead of "basal layer of the VZ"?
4. Please specify details about the statistical analysis in each experiment separately.
5. Please add quantification for Fig. S3C.
6. Quantification in Fig.4E,F should be done by normalizing the amount of cytoplasmic b-catenin to the nuclear b-catenin and not with the kap1.
7. Fig.5C-D should be moved to supplement.
8. Fig. S5E should be moved to main figures. Higher magnifications are needed. The staining of WWOX as it is in not obvious.
9. The authors write in page 16 "in the VZ of WPM F2 and WPM M3 WWOX was strongly detected", however this is not depicted in the Fig.S6E. Please provide a more representative pic. for WPM F2.

3. Significance:

Significance (Required)

The proposed study overall aims to shed light on the aetiology of a group of brain disorders called epileptic encephalopathies by scrutinizing the molecular and cellular mechanisms of the tumour suppressor and DDR gene WWOX and to contribute to the research on the molecular and cellular mechanisms that regulate brain development and which upon disruption could lead to epilepsy. This is of great importance since epilepsy is a disorder which affects a big number of people and especially children worldwide and which is known to have a developmental origin. Thus, conceptually I find the proposed work relevant considering the limited knowledge we have until now on why or how defects on human brain development may lead to such disorders. The potential contribution of the proposed work on therapeutic intervention strategies are also of great significance. To approach this the authors used brain organoids as in vitro model system which allows them to model the development of the disorder and via state-of-the-art technologies such as immunohistochemistry, electrophysiological studies and RNA sequencing approaches aim to address key questions on the development of WOREE and SCAR12 syndromes. As developmental neuroscientist working on human brain development and on modelling cortical malformation using brain organoids, I believe that such studies are necessary and crucial for understanding the development of brain related diseases and to design novel therapeutic strategies. Although major concerns regarding several experimental design strategies followed by the authors are raised as mentioned in the comments, with the proposed suggestions authors could improve their work draw more solid conclusions especially on the modelling of the WOREE syndrome using brain organoids and could contribute to the existing knowledge.

Review #2

1. How much time do you estimate the authors will need to complete the suggested revisions:

Estimated time to Complete Revisions (Required)

(Decision Recommendation)

More than 6 months

2. Evidence, reproducibility and clarity:

Evidence, reproducibility and clarity (Required)

The manuscript by Steinberg et al. examines the role of the WWOX gene in disease pathology by modeling WWOX-related epileptic encephalopathy (WOREE) syndrome using 3D brain organoids. The authors generated stable cell lines bearing WWOX mutations by CRISPR editing, or patient-derived iPSC lines. They generated cerebral organoids to study the WWOX role. They comprehensively analyzed molecular and cellular changes in cerebral organoids. Notably, they showed that organoids lacking WWOX showed defective astrogenesis, DNA damage response, neuronal populations, and differentiation. Lastly, the authors also demonstrated that patient-derived iPSC-generated forebrain organoids suffer from similar phenotypic malformations as observed in cerebral organoids. Although the findings are interesting, there are some inconsistency in the data that need further validation and replicates.

****Major comments:****

1. Authors showed that the WWOX gene plays a critical role in DNA damage. They generated WWOX knock out human pluripotent stem cell lines, but they did not give detailed description about pluripotency and genomic stability. Are these KO lines karyotypically normal? How many passages have been performed before differentiating them into neuronal lineage?

2. A number of results are not adequately quantified and thus conclusions are not fully supported. The data need quantification from enough number of repeats, considering that the cerebral organoids are highly heterogeneous. How many organoids derived from different batches used for these IF analyses, Figures 1C, 1E, 4I, and Supplementary Figures 3E, 4G and 4I? Then, please perform statistical analysis.

3. WOREE patients suffer from progressive microcephaly. Do authors observe any similar defects in the size of WWOX KO cerebral organoids compared to control organoids?

4. In Figure 2A, and 2B, authors need to explain better what the low and high frequency in field potential in brain organoids represent in vivo field potential, and how low frequency is used for EE phenotypes. Meanwhile, Figure 5 for WPM family organoids, all frequencies are low for homozygous mutant organoids. Again, there is no information for statistics, and authors need to show how many lines were used and how many repeats were made to reach the conclusion.

****Minor comments:****

1. Color labeling in Figure 2 and Supplementary Figure 2 should be presented.
2. Some figures and legends need labels. e.g. Figure 5, S3, S5. WT and mutants are not labeled (e.g. Figure 2A, B). Figure 2F and G needs legends in detail. Wrong citation of figure in the text (Figure S4H and 6C). WNT8B is not downregulated as mentioned in the text (Figure S4F).
3. The genotype in Figure S1C shows a heterozygous knock out in WKO-1B line. However, the expression of WWOX protein is not detectable in these organoids (Figure S1A). It may need explanation. Authors need to clone the PCR product and sequence each allele to see whether the clones are compound null.

3. Significance:

Significance (Required)

The manuscript shows the proof of concept in modeling Epileptic Encephalopathies (EE) using human brain organoids. Particularly, authors generated genetic model of EE using CRISPR gene editing, and measured the field potential for the neural activity.

Review #3

1. How much time do you estimate the authors will need to complete the suggested revisions:

Estimated time to Complete Revisions (Required)

(Decision Recommendation)

Between 3 and 6 months

2. Evidence, reproducibility and clarity:

Evidence, reproducibility and clarity (Required)

In their manuscript, Sandberg et al. focus on the in vitro modeling of epileptic encephalopathies (EEs) through 3D brain cultures. They evaluated the impact of WWOX KO in the cerebral organoid (CO) system by assessing alterations in neuronal differentiation, astrogliogenesis, functionality (local field potential recordings) and overall gene expression (bulk RNA-Seq). The authors report that KO-COs show upregulation of GABAergic markers, hyperexcitability, increased astrogliogenesis and DNA damage when compared to control. Among the altered molecular pathways, they emphasize the over-activation of WNT

pathway in the KO system, and validate some molecular targets of the WNT signaling with qPCR. Applying the same experimental approach, they claim that the overexpression of WWOX via CRISPR-Cas technology partially rescues the abnormalities observed in the KO. To circumvent the limits of CRISPR-edited cell models and since the majority of the phenotypes were observed in the cortical part of the CO system, they generate forebrain organoids (FOs) from two different families bearing mutations in the WWOX gene. In the "severe" patient-derived FOs the authors found alterations in astrogliogenesis, hyperexcitability and activation of the WNT pathway. The "mild" patient-derived organoids, on the other hand, showed impairment only in the WNT signaling. Apparently, this approach represents the severity of the disease at molecular level.

****Major Comments****

Results

-a SNP analyses of the WT and KO organoids should be performed to exclude that differences the authors see in their study are not also due to genetic aberrations a phenomena often seen in KO cultures due to higher passage numbers compared to controls (KO generation and expansion).

-the result that the W-AAV COs exhibit WWOX expression not only in progenitor population but also in other cell populations in which the expression could not found in controls clearly shows the ectopic overexpression of the WWOX under the UBP promotor. Thus, the authors should be very cautious when using W-AAV COs as "control" or "rescue". We highly recommend to always show and compare the analysis on the WAAV lines to the control organoids, starting from the characterization of the line in figure S1D/E. This would make the results easier to interpret and would provide more information on the impact of the overexpression. In several cases it is shown that the overexpression mitigates the effect of the KO, but it is impossible to assess to which extent this is beneficial. An example is the detrimental impact of WWOX overexpression on DNA damage/apoptosis shown in figure S3F. The authors should also clarify why they used a lentiviral infection to rescue the phenotype in the local field potential recording experiment (figure 2F) and not the W-AAV CO.

-it is very surprising that the authors describe similar levels of progenitors and neurons in the KO and WT COs and similar expression levels of vGlut but marked differences in the expression of GAD67. These data are conflicting. In case the KO and WT COs indeed show a similar level of neurons and only the KO exhibit changes in GAD67 it would be expected that there are also differences in other types of neurons not investigated. The authors showed go here into more detail as this point represents an important finding of their study. They could for instance approach this by performing single cell RNA sequencing experiments of KO and WT COs.

-It is widely known in the field that COs exhibit a high degree of variability. Thus, it is of major importance that the authors clearly indicate the number of organoids used and the number of batches the organoids were derived from in their analyses. All data should be generated using at least 3 different organoids from 3 independent batches. Showing the reproducibility of the data through different batches and across different organoids would strengthen the observation made and demonstrate the reliability of the result. These numbers

are for instance missing when the authors describe the excitability of the KO and WT COs.

-The author should also take into consideration the high heterogeneity of the CO system when discussing gene expression data. The intrinsic variability of COs could have introduced a bias, impacting the outcome of the analysis. This might be a critical point considering the authors statement: "a major part of the phenotype was observed in the cortical part of the COs". In this sense the qPCR approach on a whole CO could give a very approximative result. An example comes from the qPCRs targeting markers of cortical layering. Indeed, some of these transcripts are not exclusively expressed in the cortex, thus complicating the reading of the data. In line with this we would highly recommend to use single cell RNA seq instead of the applied whole-transcriptome RNA sequencing. A major doubt is also raised by the results of the GO enrichment showing axis specification (Ventral Dorsal & Anterior-Posterior) among the significant GO terms. The result could also be due to differences in the differentiation of the different organoid batches or even the variability within one batch.

-in panel 3C, a closer look suggests that the organoid cytoarchitecture at week 6 is obviously altered on the KO-COs although stated contrarily. We highly recommend providing overview images of the different COs derived from different batches.

-it is not clear why at page 10 the authors state that the increased DNA damage is specific for neural progenitor cells.

-Panels S3C/D and the subsequent conclusion on the origin of astroglial cells are puzzling. The authors state that increased astroglial cells in the KO-COs arise from RG cells and not proliferative APCs. Nevertheless, it is not clear how a decrease in proliferation (panel S3D) and increased differentiation of RGCs can explain the same proportion of SOX2+ cells in control and KO condition. Decreased proliferation and increased differentiation should deplete the pool of SOX2+ cells in the KO. The authors should provide a quantification of S100B/SOX2+ double positive cells. An increase in S100B progenitors could fill the gap in the assumption that differentiating RGCs are the source of astroglia.

-The authors should clarify what is the rationale behind the selection of 3000 genes in the enrichment analysis. Even though in the methods section it is stated that they are significantly modulated and ranked by log2 Fold change, the authors should define a specific cut-off for the selection.

-The authors should explain the rationale behind the normalization of b-catenin on KAP-1 (fig 4F). Although KAP-1 is used as a nuclear marker and as a proxy of the fractionation of the cytosolic and nuclear cellular compartments, a ratio between cytosolic and nuclear b-catenin levels could give a clearer picture on the nuclear translocation of the protein. This would also give a better normalization on the total levels of b-catenin.

-We highly recommend implementing a staining for TBR1 in figures 4I, S4G and S4I as a reliable cortical marker. The analysis of cortical layering markers with qPCR could be misleading due to the heterogeneity of the CO system. Moreover, as previously suggested, the data of the WAAV CO (e.g. figure S4J) should be presented together with the control COs.

-To validate the nature of the FO we suggest performing specific forebrain stainings such as EMX1 and PAX6 or TBR1 at early stages.

-In our opinion, figure 5D shows an increase in vGLUT protein levels, contradicting the statement at page 15 about a "similar expression of VGLUT1".

-In figure S6J the authors should also include a staining of the WPM M3 sample.

Discussion

-At page 17, the authors should explain the meaning of "epigenetic editing tools" in the context of their manuscript.

-At page 20, the reference to Cheng et al. 2020 is counterintuitive and against the hypothesis drawn in this manuscript. Indeed, the activation of the WNT pathway has beneficial effects on the epileptic seizures in a *Wwox*-null mice model. On the contrary, the authors associate an excessive activation of WNT pathway to the observed phenotype in the CO model. This point should be discussed in more detail.

-At page 23, we would recommend being more cautious on the therapeutic implications of these findings.

****Minor Comments****

-Please check the references to supplementary figure 4 at page 13.

-Please use the same format for the name of the markers analyzed in the captions of the figures (e.g. SOX2 in figure 1)

-Page 24 Figure 1B TUJ1: this is just the clone name of the antibody, the protein name should be always used instead (β III Tubulin)

-Page 35 "Glutamax" has been misspelled as "Gultamax".

3. Significance:

Significance (Required)

All in all, the manuscript presents a considerable amount of data obtained with an innovative approach to study brain disorders, reinforcing the use of 3D cultures as valuable in vitro tools. The study will be very interesting for a broad audience. Nevertheless, we have several concerns that the author should address.

Point-by-point reply

Reviewer #1 (Evidence, reproducibility and clarity (Required)):

In this manuscript Steinberg et al model with brain organoids the early infantile WWOX-related epileptic encephalopathy and investigate the role of WWOX in embryonic human brain development. They claim that deletion of WWOX gene has no major defects in the different neural progenitor cells but affects specifically interneurons and glial cells. They also claim that these changes are causative for the epileptiform activity observed in the KO COs while mechanistically they show that DNA damage response is affected. Interestingly, they can reverse/rescue the phenotype by re-introduction of the WWOX gene in COs. By performing RNA-seq they show that the Wnt pathway is involved in the pathophysiology of the disease. Finally they claim that patient derived organoids exhibit similar phenotype with the KO COs in terms of cellular composition, neuronal activity and DNA damage response and they suggest that these cellular and molecular differences could be important for the differences in the severity of the WOREE and SCAR12 syndromes which are both developed after mutations in the WWOX gene. Although, the authors present the data clearly and consistently, in my opinion this manuscript would benefit from a more rigorous and deep data analysis. Major concerns also are raised by the relevance of the description on the one hand of the role of WWOX in human brain development and the phenotype described in KO COs and on the other hand of the WOREE and SCAR12 syndromes and the modelling of the diseases. These concerns and further suggestions for improving this work are listed below.

Response: *We thank the reviewer for careful assessment and constructive comments that were all addressed in our revised version.*

****Major comments:****

1. Regarding the expression of WWOX in the apical side of the neuroepithelium, is it located in the apical belt or it is rather in the cytoplasm of the VZ progenitors? Co-staining with the nuclear marker SOX2 is not helpful to distinguish this. Co-staining with F-actin, b-catenin or other apical belt markers from the one hand and with Nestin from the other would be useful.

Response: *We thank the reviewer for proposing this as this is an important point. To address this quest, we stained our COs with CRYAB – a cytoplasmic protein specific for vRGs (Pollen et al., 2015, Cell). As presented in New Figure 1B, there is almost complete co-localization between CRYAB and WWOX suggesting expression of WWOX in vRGs. Description of the result appears in page 6, 1st paragraph (highlighted in yellow).*

2. The authors mention that no change in the total number of SOX2, TBR2 or NEUN were observed. However, considering the big variability between the number and size of ventricles in COs it is not clear how the authors draw these conclusions. Quantifications of the size of the VZ/CP and absolute numbers of the markers shown in Fig.1C would be useful. This will back up the RNA levels that the authors show in Fig.1F which however are inconclusive regarding the actual size of the ventricles within the COs.

Response: *We thank the reviewer for this suggestion. To address this concern, and other concerns raised by the other reviewers, we repeated the staining in two additional batches of organoids and quantify SOX2 (in the VZ and in the SVZ, as both are source for neurons and intermediate progenitors), TBR2 and NeuN. These results are presented in Fig 1C and Appendix Fig S1B. Description of the result appears in page 6, 2nd paragraph (highlighted in yellow).*

3. The authors show that in the KO COs there is big increase of GAD1 and GAD2 genes. Taking into account that these experiments were done in unpattern COs, is this correlated with general change in the identity of the cells to ventral? Are also ventral progenitor markers affected (e.g. Nkx2.1)? This could be potentially addressed by analysis of the RNA-seq data that the authors show in Fig.4

Response: *To address the inquiry of the reviewer, we analyzed the RNA seq data and as revealed by the Figure presented to the Reviewer (below), there is no significant change in dorsal genes (right) though reads are very high. Although ventral genes (left) display higher levels in KO, the overall reads are much lower compare to dorsal genes.*

Figure for reviewers removed

We have also performed qPCR for Nkx2.1 at week 15 but our results revealed no difference as well.

Therefore we conclude that our COs are majorly of dorsal identity.

Figure for reviewers removed

4. The authors mention in the introduction that WOREE-patients may have microcephaly. It would be interesting if the authors check whether there are any changes in the total size of the COs either in the KO or in the patient-derived FOs.

Response: *This is an interesting point- although microcephaly is considered a part of the WOREE syndrome, a review article from 2018 claimed only 30% of cases present with microcephaly (Piard et al., 2019, Genetics in Medicine). This was described in the introduction, end of page 3 (highlighted in yellow).*

Furthermore, in case of the family referred in this paper as WSM, which harbor, out of 2 affected children with the same mutation, only one was clearly defined as microcephalic, with the patients whose iPSCs are used in this study, described with normal head circumference (3rd percentile) (Weisz-Hubshman et al., 2019, EJPN). This phenomenon, of patients with the same genotype presenting with a bit different phenotype is quite common. In regard to the WPM family, microcephaly has not been described (Gribaa et al., 2007, Brain; Mallaret et al., 2014, Brain).

5. The rescue of the phenotype observed from the W-AAV COs is interesting. However, more detailed analysis needs to be done comparing the neural markers expression of W-AAV with the ctrl COs. This would be important since the authors propose the use of W-AAV as a rescue strategy. It would be also important to include in this analysis quantifications of the protein levels of the neural markers by stainings or by western blot.

Response: *We agree with the reviewer that this is as an important point. To address this, we quantified immunostaining of week 10 COs as presented in Fig 1C, highlighted in yellow in page 6, 2nd paragraph. Essentially the outcome of this revealed comparable numbers of radial glia (SOX2), intermediate progenitors (TBR2) and neurons (NeuN).*

Figure for reviewers removed

We have also quantified GFAP using WB analysis at week 20 W-AAV COs compared to WT (shown here) and compared week 24 WT to KO (Figure EV2A, B). Our immunoblot analysis supports the immunostaining (Figure 3A).

6. It is unclear how/why the authors conclude that the reduction of the proliferating rate of SOX2+ progenitors lead to increase of astrocytes as mentioned in the last sentence of page 9. The increased differentiation of SOX2+ cells may lead to increased neuronal differentiation since authors show also increased neuronal markers in Fig.1. Authors may think of electroporating the KO COs with GFP or to KD the WWOX gene in ctrl organoids via electroporation and follow their progeny. This is an important experiment

in my opinion.

Response: We thank the reviewer for this suggestion. We attempted to perform electroporation for GFP vector but this experiment was unsuccessful as number of GFP-positive cells was very low. We hence tried to address the reviewer's inquiry by performing BrdU labeling (data presented below). The assumption was that after additional 2 weeks of culture, the BrdU will "fade out" of the cells that proliferate and persist in the cells that directly convert into a mature form. Therefore, if the astrocytes arise from the RG cells in the KO, a higher proportion of BrdU⁺/S100β⁺ will be seen compared to the WT. To this end, we treated WT and KO week-4 brain organoids with BrdU, 48 hours later the media was washed and organoids were collected 2-weeks later for immunostaining as shown below. The outcome of this experiment was inconclusive. Although we observed that 26% of the BrdU labelled cells in the KO organoids were S100β-positive and 18% in the WT organoids, this was statistically insignificant. NeuN-positive cells that are labelled with BrdU were in similar proportion in WT (34%) and KO (31%). It is important to note that the WT organoids exhibited higher proportion of BrdU labelled nuclei relative to overall nuclei as compared to KO. We believe this experiment should be considered as preliminary and that an in-depth analysis should be done in the future to specifically address this issue.

Figures for reviewers removed

7. The finding with increased DDR is very interesting. Is this increase linked with apoptosis as the authors imply? Caspase-3 or tunnel staining would be useful. If so, how the authors explain the similar numbers of SOX2+ progenitors in the KO COs?

Response: As we find this question very relevant, we performed IF staining for cleaved caspase 3 and focused on its expression in the VZ. As expected, we found decreased expression in our WWOX-KO COs compared to WT COs, which was reversed in the W-AAV COs. The images are presented in Figure EV2G and described in page 10, end of 2nd paragraph (highlighted in yellow). As noted in the discussion in page 20, we propose that WWOX KO vRGs accumulate DNA breaks that could "direct-convert" to astrocytic-like cells.

8. One major concern regarding the modelling of the WOREE syndrome and the role of WWOX in this syndrome and the human cortical formation is the fact that the patient-derived iPSCs from the sick son (WSMS5) have similar WWOX protein levels with the

healthy father and these levels are very much reduced compared to wt (Fig.5A). Should the authors consider using at least the WSMM2 which resembles closely the wt situation? The same situation is observed also in the second family WPM. Fig. S5A doesn't include wt iPSCs while the quality of the pictures of the western blot in the D1 and S1 is bad quality. If the protein is present how can the authors correlate the phenotype observed with the KO COs? If the authors believe that the WWOX in the patients is not functional this needs to be addressed.

Response: *We thank the reviewer for this suggestion. To address this inquiry, we generated a new batch of COs of the WSM family (including WSM M2 and WSM S W-AAV) that is presented in New figure 5 & 6 and are described in pages 14-16. In this new batch of COs we confirmed the results observed in hESCs-derived COs and those obtained using the FO protocol (Appendix Figure S2-3). Quality of the WB was improved by re-running, shown in Appendix Figure S2 and S4.*

9. Another major concern regarding experimental design of the study: It is unclear why the authors decide to generate forebrain organoids when using the patients-derived iPSCs and not continue with the COs since their aim is to compare the patients' phenotype with the KO. And if they want to generate more fate driven organoids why they do not choose the dorsal and ventral organoids published from (Pasca and Knoblich labs) since they do see major differences in the inhibitory vs excitatory neurons (Fig.1). In my opinion the authors should include at least part of the analysis in the unpattern organoids and/or in ventral and dorsal assembloids/fused organoids.

Response: *We thank the reviewer for this inquiry. Per request of the Reviewer and as mentioned in our previous response (#8), we generated a new batch of COs of the WSM family that is presented in New figure 5 & 6 and are described in pages 14-16. In this new batch of COs we confirmed the results observed in hESCs-derived COs and those obtained using the FO protocol (Appendix Figure S2-3).*

10. Contrast to what authors claim, VGlut1 doesn't seem similar between the affected son and the unaffected father in Fig.5D. In addition, GAD67 staining is not clear from the pictures. Quantifications and a more representative pictures and magnifications are needed.

Response: *We agree with the reviewer that the images were confusing. We performed additional staining on the original batch and the new batch of WSM COs and quantified the staining. Data are presented in Fig 1E & F [page 7, 1st paragraph] and New Figure 5D & E [described in page 16, 2nd paragraph].*

11. It is not clear why the authors claim that the SCAR12 individuals do not show differences in the astrocytes and DDR. According to Fig.S6I,J,K the WPMS1 shows

increased astrocytic markers (from quantifications, GFAP and ALDH1A1) and increased 53BP1 and γ H2AX (staining). This patient also has the lowest WWOX expression according to Fig.S6A. In my view this is the patient who resembles the most the KO situation. Thus, the conclusions made from the authors are contradicting.

Response: *Although our interpretation of the data differs from that of the reviewer's, we do appreciate the note and therefore performed new staining on additional sections and added WPM M3 (mother) in the new analysis (Figure EV5A). Additionally, we are showing that the levels of astrocytic markers are not showing consistent increase in the patients, as assessed by qPCR (Figure EV5B).*

****Minor comments:****

1. Discuss previous publications on the role of WWOX in human brain and human neural progenitor cells. Special emphasis should be given to the comparison of the DEGs in these publications and in the present manuscript.

Response: *As requested, we discussed two recent papers highlighting the effects of WWOX-silencing in 2D human neuronal progenitor cells (hNPCs), with one of them re-analyzing the data of the earlier one, therefore describing similar results. This is described in page 22 (highlighted in yellow).*

2. Show WWOX protein levels in the W-AAV COs compared to ctrl levels in Fig S1D.

Response: *As requested, we are showing this by immunostaining in Figure 1A and Western blot presented in response to comment # 5.*

3. Page 7 in the paragraph: "These findings suggest that during human embryonic....." are the authors mean "apical layer of the VZ" instead of "basal layer of the VZ"?

Response: *Yes, we thank the reviewer for the keen observation.*

4. Please specify details about the statistical analysis in each experiment separately.

Response: *The details were added to the legend of each figure, as requested.*

5. Please add quantification for Fig. S3C.

Response: *A misprint occurred in the legend, and figure S3C is quantified in figure S3D. In the current version, this is presented as Figure EV2C, D.*

6. Quantification in Fig.4E,F should be done by normalizing the amount of cytoplasmic b-catenin to the nuclear b-catenin and not with the kap1.

Response: *The requested comparison was performed and added to the figure (Figure 4E).*

7. Fig.5C-D should be moved to supplement.

Response: As requested, we moved this data to supplement, presented as Appendix Figure S3B, C.

8. Fig. S5E should be moved to main figures. Higher magnifications are needed. The staining of WWOX as it is in not obvious.

Response: Figure S5E (currently Appendix Figure S3H) is showing qPCR results for Wnt related genes. We assume that the reviewer referred to the original figure S5D which is now reproduced as has been now reproduced in Appendix Figure S2D in higher magnification. As we have no space in the main figures, we propose to keep this in the appendix.

9. The authors write in page 16 "in the VZ of WPM F2 and WPM M3 WWOX was strongly detected", however this is not depicted in the Fig.S6E. Please provide a more representative pic. for WPM F2.

Response: As requested, we provided a new representative image, new Figure EV4A. It is interesting to note that WPM F2 expresses lower WWOX levels compared to WPM M3 (WPM M3 is 1.8 fold higher), and therefore it is not visualized at the same levels.

Reviewer #1 (Significance (Required)):

The proposed study overall aims to shed light on the aetiology of a group of brain disorders called epileptic encephalopathies by scrutinizing the molecular and cellular mechanisms of the tumour suppressor and DDR gene WWOX and to contribute to the research on the molecular and cellular mechanisms that regulate brain development and which upon disruption could lead to epilepsy. This is of great importance since epilepsy is a disorder which affects a big number of people and especially children worldwide and which is known to have a developmental origin. Thus, conceptually I find the proposed work relevant considering the limited knowledge we have until now on why or how defects on human brain development may lead to such disorders. The potential contribution of the proposed work on therapeutic intervention strategies are also of great significance. To approach this the authors used brain organoids as in vitro model system which allows them to model the development of the disorder and via state-of-the-art technologies such as immunohistochemistry, electrophysiological studies and RNA sequencing approaches aim to address key questions on the development of WOREE and SCAR12 syndromes. As developmental neuroscientist working on human brain development and on modelling cortical malformation using brain organoids, I believe that such studies are necessary and crucial for understanding the development of brain related diseases and to design novel therapeutic strategies. Although major concerns regarding several experimental design strategies followed by the authors are raised as mentioned in the comments, with the proposed suggestions authors could improve their work draw more solid conclusions especially on the modelling of the WOREE syndrome using brain organoids and could contribute to the existing knowledge.

Response: We very much appreciate the statement of the Reviewer and thank him/her

for acknowledging the significance of our paper. We hope we successfully addressed all the proposed suggestions as described in our revised manuscript.

Reviewer #2 (Evidence, reproducibility and clarity (Required)):

The manuscript by Steinberg et al. examines the role of the WWOX gene in disease pathology by modeling WWOX-related epileptic encephalopathy (WOREE) syndrome using 3D brain organoids. The authors generated stable cell lines bearing WWOX mutations by CRISPR editing, or patient-derived iPSC lines. They generated cerebral organoids to study the WWOX role. They comprehensively analyzed molecular and cellular changes in cerebral organoids. Notably, they showed that organoids lacking WWOX showed defective astrogenesis, DNA damage response, neuronal populations, and differentiation. Lastly, the authors also demonstrated that patient-derived iPSC-generated forebrain organoids suffer from similar phenotypic malformations as observed in cerebral organoids. Although the findings are interesting, there are some inconsistency in the data that need further validation and replicates.

Response: *We thank the reviewer for the summary and finding our results interesting. We addressed all concerns as detailed below.*

.
****Major comments:****

1. Authors showed that the WWOX gene plays a critical role in DNA damage. They generated WWOX knock out human pluripotent stem cell lines, but they did not give detailed description about pluripotency and genomic stability. Are these KO lines karyotypically normal? How many passages have been performed before differentiating them into neuronal lineage?

Response: *We thank the reviewer for this valid concern. We performed karyotype analysis on WT hESCs and both KO clones, presented here for the reviewer. As seen below, we obtained normal karyotype of both genotypes. We have also injected these lines into NOD/SCID mice and obtained teratomas as also shown below.*

Figures for reviewers removed

2. A number of results are not adequately quantified and thus conclusions are not fully supported. The data need quantification from enough number of repeats, considering that the cerebral organoids are highly heterogeneous. How many organoids derived from different batches used for these IF analyses, Figures 1C, 1E, 4I, and Supplementary Figures 3E, 4G and 4I? Then, please perform statistical analysis.

Response: *We thank the reviewer for this important comment. We have performed quantification on all requested from at least three different batches for most of the data for the referred figures, currently are indicated as Figure 1C, 1F, 4H, 3F and EV2F. In our revised manuscript, we referred to this in each figure legend and indicated the statistical analysis performed.*

3. WOREE patients suffer from progressive microcephaly. Do authors observe any similar defects in the size of WWOX KO cerebral organoids compared to control organoids?

Response: *This is an interesting point- although microcephaly is considered a part of the WOREE syndrome, a review from 2018 claimed only 30% of cases present with microcephaly (Piard et al., 2019, Genetics in Medicine). This was described in the introduction, end of page 3 (highlighted in yellow).*

Furthermore, in case of the family referred in this paper as WSM, which harbor a, out of 2 affected children with the same mutation, only one was clearly defined as microcephalic, with the patients whose iPSCs are used in this study, described with normal head circumference (3rd percentile) (Weisz-Hubshman et al., 2019, EJPN). This phenomenon, of patients with the same genotype presenting with a bit different phenotype is quite common. In regard to the WPM family, microcephaly has not been described (Gribaa et al., 2007, Brain; Mallaret et al., 2014, Brain).

4. In Figure 2A, and 2B, authors need to explain better what the low and high frequency in field potential in brain organoids represent in vivo field potential, and how low frequency is used for EE phenotypes. Meanwhile, Figure 5 for WPM family organoids, all frequencies are low for homozygous mutant organoids. Again, there is no information for statistics, and authors need to show how many lines were used and how many repeats were made to reach the conclusion.

Response: *We thank the reviewer for this suggestion. We have now explained better the description and added an extended discussion in pages 18-20 (highlighted in yellow). We have also performed a new batch of COs for the WSM family and conducted cell-attached recordings that appear in New Figure 5B, C and are described in page 15, 2nd paragraph. All statistical analysis are described in Figure legends.*

****Minor comments:****

1. Color labeling in Figure 2 and Supplementary Figure 2 should be presented.

Response: Color labeling was added in all the requested figures, currently appearing as Figure 2 and EVF1G, H.

2. Some figures and legends need labels. e.g. Figure 5, S3, S5. WT and mutants are not labeled (e.g. Figure 2A, B). Figure 2F and G needs legends in detail. Wrong citation of figure in the text (Figure S4H and 6C). WNT8B is not downregulated as mentioned in the text (Figure S4F).

Response: Labels were added to all of the requested figures. Legends were expanded and mistakes were fixed as requested.

3. The genotype in Figure S1C shows a heterozygous knock out in WKO-1B line. However, the expression of WWOX protein is not detectable in these organoids (Figure S1A). It may need explanation. Authors need to clone the PCR product and sequence each allele to see whether the clones are compound null.

Response: This is an important note that was not clarified due to space limitations – as seen in figure S1A, one allele of the mutation harbors an insertion of 1 T nucleotide, while the other allele harbors an insertion of 4 T nucleotides – this eventually causes the same downstream premature stop codon. If required, we will add more details regarding the consequences of these insertions.

Reviewer #2 (Significance (Required)):

The manuscript shows the proof of concept in modeling Epileptic Encephalopathies (EE) using human brain organoids. Particularly, authors generated genetic model of EE using CRISPR gene editing, and measured the field potential for the neural activity.

Response: We thank the Reviewer for acknowledging the significance of our paper.

Reviewer #3 (Evidence, reproducibility and clarity (Required)):

In their manuscript, Sandberg et al. focus on the in vitro modeling of epileptic encephalopathies (EEs) through 3D brain cultures. They evaluated the impact of WWOX KO in the cerebral organoid (CO) system by assessing alterations in neuronal differentiation, astrogliogenesis, functionality (local field potential recordings) and overall gene expression (bulk RNA-Seq). The authors report that KO-COs show upregulation of GABAergic markers, hyperexcitability, increased astrogliogenesis and DNA damage when compared to control. Among the altered molecular pathways, they emphasize the over-activation of WNT pathway in the KO system, and validate some molecular targets of the WNT signaling with qPCR. Applying the same experimental approach, they claim that the overexpression of WWOX via CRISPR-Cas technology partially rescues the abnormalities observed in the KO. *To circumvent the limits of CRISPR-edited cell models and since the majority of the phenotypes were observed in the cortical part of the CO system, they generate forebrain organoids (FOs) from two different families bearing mutations in the WWOX gene.* In the "severe" patient-derived FOs the authors found alterations in astrogliogenesis, hyperexcitability and activation of the WNT pathway. The "mild" patient-derived organoids, on the other hand, showed impairment only in the WNT signaling. Apparently, this approach represents the severity of the disease at molecular level.

****Major Comments****

Results

1. a SNP analyses of the WT and KO organoids should be performed to exclude that differences the authors see in their study are not also due to genetic aberrations a phenomena often seen in KO cultures due to higher passage numbers compared to controls (KO generation and expansion).

Response: *We thank the reviewer for this valid concern. We performed karyotype analysis on WT hESCs and both KO clones, presented in Response to **Reviewer 2, comment #1**. As shown, we obtained normal karyotype of both genotypes. We have also injected these lines into NOD/SCID mice and obtained teratomas as also shown.*

2. the result that the W-AAV COs exhibit WWOX expression not only in progenitor population but also in other cell populations in which the expression could not found in controls clearly shows the ectopic overexpression of the WWOX under the UBP promotor. Thus, the authors should be very cautious when using W-AAV COs as "control" or "rescue". We highly recommend to always show and compare the analysis on the WAAV lines to the control organoids, starting from the characterization of the line in figure S1D/E. This would make the results easier to interpret and would provide more information on the impact of the overexpression. In several cases it is shown that the

overexpression mitigates the effect of the KO, but it is impossible to assess to which extent this is beneficial. An example is the detrimental impact of WWOX overexpression on DNA damage/apoptosis shown in figure S3F. The authors should also clarify why they used a lentiviral infection to rescue the phenotype in the local field potential recording experiment (figure 2F) and not the W-AAV CO.

Response: *We thank the reviewer for this important suggestion. We have added the W-AAV results in the different assays and Figures comparing WT/KO/W-AAV in the new revised manuscript. We have also performed a new batch of COs on the WSM family (including W-AAV rescue in WSM S) that is presented as New Figure 5 & 6 and included electrophysiology recording. As presented, W-AAV rescued most of the phenotypes, as assessed by the various methods used. We added in the discussion a note addressing this, page 23, 2nd paragraph (highlighted in yellow). These results are supported with our initial attempts when we used the available Lenti-WWOX vector. Our data with W-AAV are reproducible, very encouraging and are considered proof-of-concept as we emphasized in our paper. Future work will be required to optimize population-targeting delivery and fine tuning of expression levels for successful genetic therapy approaches in WOREE patients.*

3. it is very surprising that the authors describe similar levels of progenitors and neurons in the KO and WT COs and similar expression levels of vGlut but marked differences in the expression of GAD67. These data are conflicting. In case the KO and WT COs indeed show a similar level of neurons and only the KO exhibit changes in GAD67 it would be expected that there are also differences in other types of neurons not investigated. The authors showed go here into more detail as this point represents an important finding of their study. They could for instance approach this by performing single cell RNA sequencing experiments of KO and WT COs.

Response: *We agree with the reviewer that this is an intriguing point to address. However, we think that performing scRNA-seq analysis would be beyond the scope of this current manuscript. Instead, we repeated the staining in additional batches of organoids and quantified the different markers of neurons. These results are presented in Fig 1D-F. In the new batch of COs of WSM family, we have also quantified GAD67 and vGlut1 which is presented in Fig 5D, E and observed consistent data with hESCs-derived COs. Additionally, we quantified levels of SOX2 in the VZ (Fig 1C and Appendix Fig S1B) and SVZ (Figure EV2D) and observed no difference. These data are also consistent with total protein levels for SOX2 and NeuN, as assessed by Western blotting (Figure EV2A, B). All together, these findings support our original observations. Future analysis using scRNA-seq analysis will be certainly help further explain this intriguing outcome.*

4. It is widely known in the field that COs exhibit a high degree of variability. Thus, it is

of major importance that the authors clearly indicate the number of organoids used and the number of batches the organoids were derived from in their analyses. All data should be generated using at least 3 different organoids from 3 independent batches. Showing the reproducibility of the data through different batches and across different organoids would strengthen the observation made and demonstrate the reliability of the result. These numbers are for instance missing when the authors describe the excitability of the KO and WT COs.

Response: *We totally agree with the reviewer that the heterogeneity of COs is widely known. This was addressed in our manuscript by using another protocol of patterned organoids (forebrain organoids) and we reached comparable results and outcomes. Additionally, as stated above we generated a new batch of COs for the WSM family and obtained similar results compared to Forebrain organoids and hESCs-derived COs; data presented in New Figure 5 and 6 and are described in pages 14-16. Quantifications and conclusions were made using at least three batches for most of the data, as indicated in Figure legends.*

5. The author should also take into consideration the high heterogeneity of the CO system when discussing gene expression data. The intrinsic variability of COs could have introduced a bias, impacting the outcome of the analysis. This might be a critical point considering the authors statement: "a major part of the phenotype was observed in the cortical part of the COs". In this sense the qPCR approach on a whole CO could give a very approximative result. An example comes from the qPCRs targeting markers of cortical layering. Indeed, some of these transcripts are not exclusively expressed in the cortex, thus complicating the reading of the data. In line with this we would highly recommend to use single cell RNA seq instead of the applied whole-transcriptome RNA sequencing. A major doubt is also raised by the results of the GO enrichment showing axis specification (Ventral Dorsal & Anterior-Posterior) among the significant GO terms. The result could also be due to differences in the differentiation of the different organoid batches or even the variability within one batch.

Response: *We agree with the reviewer that doing scRNA-seq will be informative but we again believe this is beyond the scope of this particular study. It should be noted that the qPCR data we are presenting was set to support the presented immunostaining data, which allows, to a great extent, distinguish between cell types and areas. Our conclusions in CRISPR-edited hESCs were further validated on iPSCs from patients and using different protocols (COs and FOs). Furthermore, the observation of changes in axis specification led us to examine the changes in the Wnt pathway, which revealed activation in KO COs. For example, when we examined Wnt genes over time (Figure EV3C), since our RNA isolation technique does not allow the re-use of an organoids, we performed the experiment in at least 3 organoids in each time point from overall 3 batches. Additionally, the subfractionation experiment was done on a totally different batch of the one we performed the RNA-seq. The fact that the activation was still observed reduces the chance this is a result of intra-batch variability.*

6. in panel 3C, a closer look suggests that the organoid cytoarchitecture at week 6 is obviously altered on the KO-COs although stated contrarily. We highly recommend providing overview images of the different COs derived from different batches.

Response: *We provide here an overview picture of the different organoids stained with H&E showing similar cytoarchitecture.*

Figure for reviewers removed

7. it is not clear why at page 10 the authors state that the increased DNA damage is specific for neural progenitor cells.

Response: *We apologize for not clarifying this important point. We do not think the damage is specific, but we decided to focus on the WWOX-expressing cells in the organoids (vRGs). In fact, our preliminary data, which was not shown here, suggests the damage is maintained in other cell types as well. We referred to this in the manuscript, page 20, highlighted in yellow.*

8. Panels S3C/D and the subsequent conclusion on the origin of astroglial cells are puzzling. The authors state that increased astroglial cells in the KO-COs arise from RG cells and not proliferative APCs. Nevertheless, it is not clear how a decrease in proliferation (panel S3D) and increased differentiation of RGCs can explain the same proportion of SOX2+ cells in control and KO condition. Decreased proliferation and increased differentiation should deplete the pool of SOX2+ cells in the KO. The authors should provide a quantification of S100B/SOX2+ double positive cells. An increase in S100B progenitors could fill the gap in the assumption that differentiating RGCs are the source of astroglia.

Response: *We agree with the reviewer that a clearer explanation should be provided, and we appreciate raising this valid question. To our knowledge, the source of astrocytes during development is still largely unknown, and astrocytes are postulated to*

develop through several different pathways. Such pathways can be proliferation of mature radial glia, direct transformation, through asymmetric divisions (that generates intermediate progenitors and conserves the RGs pool) and etc. Our main claim is that vRGs, which give rise to both neurons and astrocytes, upon loss of WWOX increase astrocytes production, without claiming for a certain pathway.

The suggestion that the main source of astrocytes is the vRGs is based mainly on 4 observations:

- 1) As depicted in fig EV1C, we did not observe an increase in S100B+/KI67+ cells.*
- 2) All of the Ki67+ cells observed were also SOX2+ cells, but when normalizing KI67+ positive cells to SOX2+ outside the VZ, a decrease in the proliferating fraction was seen, suggesting the increased proliferation is not stemming from oSVZ/CP areas. These Ki67+ cells could include proliferating oRGs, asymmetrical division of oRGs, proliferation of APCs and etc. This does not exclude proliferation of mature astrocytes, but we did not observe it, which is in line with their known low proliferation rate.*
- 3) WWOX is mainly expressed at this stage by vRGs, and therefore they are very likely to be affected by its loss, compare to cells outside the VZ which do not express it.*
- 4) As mentioned in the discussion, previous research suggests that NSCs that accumulate DNA damage acquire astrocytic-like characteristics.*

To better prove this point of increased astrocytes from RG origin, we attempted to label the RGs with BrdU before the emergence of astrocytes (week 4 of development, when most of the cells are RGs). The assumption was that after additional 2 weeks of culture, the BrdU will “fade out” of the cells that proliferate and persist in the cells that directly convert into a mature form. Therefore, if the astrocytes arise from the RG cells in the KO, a higher proportion of BrdU+/S100 β + will be seen compared to the WT.

The outcome of this experiment was inconclusive (shown below). Although we observed that 26% of the BrdU labelled cells in the KO organoids were S100 β -positive and 18% in the WT organoids, this was statistically insignificant. NeuN-positive cells that are labelled with BrdU were in similar proportion in WT (34%) and KO (31%). It is important to note that the WT organoids exhibited higher proportion of BrdU labelled nuclei relative to overall nuclei as compared to KO. We believe this experiment should be considered as preliminary and that an in-depth analysis should be done in the future to specifically address this issue.

Figures for reviewers removed

Regarding the SOX2⁺ pool – in general, we did not see a decrease in progenitor pool (new fig 1C and fig EV2C-D). Outside the VZ we observed less proliferation (fig EV2C-D) but we did not claim for enhanced differentiation of these cells, therefore it is logical there is no change in the progenitor pool.

Inside the VZ, we observed decreased apoptosis and increased number of proliferating damaged cells (fig EV2E-G) – this observation can suggest the progenitor pool should increase in size! Since we did not observe this as well, increased differentiation could explain this.

In conclusion, considering the data in the manuscript, and together with the preliminary data from the BrdU experiment, we propose a model in which upon loss of WWOX, vRGs accumulate DNA damage, but since they cannot go through apoptosis, they instead differentiate into astrocytic-like cells as a mean of regulation. More future research is needed to prove this hypothesis.

9. The authors should clarify what is the rationale behind the selection of 3000 genes in the enrichment analysis. Even though in the methods section it is stated that they are significantly modulated and ranked by log₂ Fold change, the authors should define a specific cut-off for the selection.

Response: *We apologize for not explaining the rationale clearly enough. The selection of 3000 genes corresponds with DEG significance p-value threshold of 0.02. We have also tested a tighter set of 1000 genes corresponding to p-value threshold of 1E-3. So the decision was determined (made empirically), after examining a few cut-off options. If required, we will attach a few more options to the manuscript. This was also added to the manuscript under the methods section (page 36).*

10. The authors should explain the rationale behind the normalization of b-catenin on KAP-1 (fig 4F). Although KAP-1 is used as a nuclear marker and as a proxy of the fractionation of the cytosolic and nuclear cellular compartments, a ratio between cytosolic and nuclear b-catenin levels could give a clearer picture on the nuclear translocation of the protein. This would also give a better normalization on the total levels of b-catenin.

Response: *We kindly thank the reviewer for his note. First, a quantification of cytosolic β -catenin was added together with normalization of nuclear β -catenin to it (new fig 4E). We still believe a normalization to KAP-1 should be presented, as mean to normalize to the amount of protein originating specifically from the nucleus. Altogether, we believe that these two analyses complement each-other and give a more accurate view of the data.*

11. We highly recommend implementing a staining for TBR1 in figures 4I, S4G and S4I as a reliable cortical marker. The analysis of cortical layering markers with qPCR could be misleading due to the heterogeneity of the CO system. Moreover, as previously suggested, the data of the WAAV CO (e.g. figure S4J) should be presented together with the control COs.

Response: We thank the reviewer for the suggestion. As requested, we grouped together WT, KO and W-AAV COs, stained for TBR1 in multiple batches (Fig 4G and fig EV3E) and quantified the data (Fig 4H and Fig 6F).

12. To validate the nature of the FO we suggest performing specific forebrain stainings such as EMX1 and PAX6 or TBR1 at early stages.

Response: As this is an important note in our eyes, we performed a staining for PAX6 early in our FOs development to confirm dorsal identity of the progenitor cells (Appendix fig S4E).

13. In our opinion, figure 5D shows an increase in vGLUT protein levels, contradicting the statement at page 15 about a "similar expression of VGLUT1".

Response: We agree with the reviewer that the images were confusing. Therefore, we repeated the staining in a new batch of WSM COs and quantified it (fig 6D, E).

14. In figure S6J the authors should also include a staining of the WPM M3 sample.

Response: We agree with the reviewer and the staining is now updated in fig EV5A.

Discussion

15. At page 17, the authors should explain the meaning of "epigenetic editing tools" in the context of their manuscript.

Response: We referred by that to "reprogramming", but we do see the problem and the phrase was replaced (page 18, highlighted in yellow).

16. At page 20, the reference to Cheng et al. 2020 is counterintuitive and against the hypothesis drawn in this manuscript. Indeed, the activation of the WNT pathway has beneficial effects on the epileptic seizures in a *Wwox*-null mice model. On the contrary, the authors associate an excessive activation of WNT pathway to the observed phenotype in the CO model. This point should be discussed in more detail.

Response: We apologize for the confusion this discussion point caused. Chen et al (2020) discuss the activation of GSK-3 β (in theory, inhibition of Wnt) and its inhibition by lithium (allowing, hypothetically, for activation of WNT), but they did not address Wnt-status in the adult rodents. As we refer to human, embryonic organoids, and our data suggest activation of WNT, this reference, although very important, is confusing. We therefore removed the reference on hope this makes the discussion easily understood.

17. At page 23, we would recommend being more cautious on the therapeutic implications of these findings.

Response: We agree and attempted to be more cautious. As we are now showing in both hESCs-derived COs and in patient-derived COs, the vast majority of the

phenotypes are reversible through WWOX re-introduction. This of course does not mean all phenotypes, therefore we attempt to better emphasize this was only a partial rescue, as now mentioned in page 23 (highlighted in yellow).

****Minor Comments****

1. Please check the references to supplementary figure 4 at page 13.

Response: *We apologize for this mistake. While working on the different versions of the manuscript with our collaborators a mistake occurred and is now fixed. We thank the reviewer for the keen eye and noticing this mistake.*

2. Please use the same format for the name of the markers analyzed in the captions of the figures (e.g. SOX2 in figure 1)

Response: *All the requested changes will be made.*

3. Page 24 Figure 1B TUJ1: this is just the clone name of the antibody, the protein name should be always used instead (β III Tubulin)

Response: *All TUJ1 mentions will be replaced by β 3-Tubulin.*

4. Page 35 "Glutamax" has been misspelled as "Gultamax".

Response: *We again thank the reviewer for noticing these easy-to-miss mistakes and improving the quality of the writing.*

Reviewer #3 (Significance (Required)):

All in all, the manuscript presents a considerable amount of data obtained with an innovative approach to study brain disorders, reinforcing the use of 3D cultures as valuable in vitro tools. The study will be very interesting for a broad audience. Nevertheless, we have several concerns that the author should address.

Response: *We very much appreciate the statement of the Reviewer and thank him/her for acknowledging the significance of our paper. We will address all the proposed suggestions as described in our revision plan.*

the LAUTENBERG
CENTER *for General and Tumor
Immunology at the Hebrew
University of Jerusalem*

מרכז ע"ש לאוטנברג לאימונולוגיה כללית ולאימונולוגיה של גידולים

April 22, 2021

Dr. Zeljko Durdevic

Scientific Editor

EMBO Molecular Medicine

z.durdevic@embomolmed.org

Dear Dr. Durdevic,

We are happy and excited to submit our revised manuscript to EMBO Molecular Medicine for manuscript# EMM-2020-13610 [RC-2020-00431] and entitled "Modeling Genetic Epileptic Encephalopathies using Brain Organoids". We would like to thank you and the reviewers for the constructive reviews. We addressed all comments and suggestions as requested [highlighted in yellow] and we feel this greatly improved our manuscript. A-point-by-point explanation is enclosed but I would like here to emphasize the key changes we have made considering your decision letter on Oct 22, 2020.

Referee #1

1. All the main findings were reproduced in at least 3-batches of organoids and this was indicated in the Figure Legend. It is up to the Editor to keep this information in the Legend as it took too much space. We have also generated a new batch of patient-derived cortical organoids (COs) as requested and were able to reproduce all the data originally reported in patient-derived forebrain organoids and hESC-edited COs. The data appear in New Figure 5 & 6 and are described in pages 14-16.
2. To confirm WWOX expression vRGs, we co-stained with CRYAB – a cytoplasmic protein specific. Data are presented in New Figure 1B, showing almost complete co-localization between CRYAB and WWOX suggesting expression of WWOX in vRGs. Description of the result appears in page 6, 1st paragraph (highlighted in yellow).

3. We added a statement regarding microcephaly stating that although it is considered a part of the WOREE syndrome, a review article from 2018 claimed only 30% of cases present with microcephaly (Piard et al., 2019, Genetics in Medicine). This was described in the introduction, end of page 3 (highlighted in yellow).
Furthermore, in case of the family referred in our paper as WSM, which harbor, out of 2 affected children with the same mutation, only one was clearly defined as microcephalic, with the patients whose iPSCs are used in this study, described with normal head circumference (3rd percentile) (Weisz-Hubshman et al., 2019, EJPN). This phenomenon, of patients with the same genotype presenting with a bit different phenotype is quite common. In regard to the WPM family, microcephaly has not been described (Gribaa et al., 2007, Brain; Mallaret et al., 2014, Brain).

Referee #3

4. As instructed we didn't perform scRNA-seq. A detailed/extended analyses using qPCR, immunostaining, Western blotting has been performed as requested.
Furthermore, quantification was done and is shown all through the manuscript for neuronal markers (Fig 1C, Fig 1F, Fig 5E), cortical layering (Fig 6F) and DNA damage (Fig 3F, Fig 6B, Fig EV2F, Fig EV5D, Fig S3F). Immunostaining was done on at least 3-batches for the most part and are indicated in the Figure Legend. It is up to the Editor to keep this information in the Legend as it took too much space.
5. Per request of the Editor and reviewers, we have generated a new batch of patient-derived cortical organoids (COs) as requested and were able to reproduce all the data originally reported in patient-derived forebrain organoids and hESC-edited COs. Of particular interest is the rescue we observed when analyzing the W-AAV organoids showing very promising results that can be considered as proof-of-concept for replacement therapy in WOREE syndrome. The data appear in New Figure 5 & 6 and are described in pages 14-16.

The extent of the work required us to involve new authorships that are updated in the revised version. Four authors: Irina Kustanovich, Sergey Viukov, Baraa Abudiab and Shani Stern were added.

We believe our current paper is highly improved and would be greatly valued by the broad spectrum of EMBO Molecular Medicine scientific community.

We thank you in advance for your consideration and looking forward to hearing from you.

Sincerely yours,

Rami I. Aqeilan, PhD

Chair, Institute for Medical Research Israel-Canada (IMRIC)

11th May 2021

Dear Prof. Aqeilan,

Thank you for the submission of your revised manuscript to EMBO Molecular Medicine. We have now heard back from the three referees who we asked to re-evaluate your manuscript. As you will see from the reports below, the referees are overall supporting publication of your manuscript but also raise some concerns that should be addressed in an additional and final round of revision. Please address all the points raised by the referees and implement all suggested adjustments. We would like to encourage you to address referee #1 points 3 and 9 experimentally, however, this is not essential and can be also addressed in writing. Furthermore, we noticed that Fig EV5D contains a boxplot with 'WPM F2: n=1'; please revise and add an additional sample or consider an alternative way to visualize the quantification.

Acceptance or rejection of the manuscript will depend on the completeness of your responses included in the next, final version of the manuscript. For this reason, and to save you from any frustrations in the end, I would strongly advise against returning an incomplete revision.

In addition, please amend the following:

1) In the main manuscript file, please do the following:

- Correct/answer the track changes suggested by our data editors by working from the attached/uploaded document.
- Reduce number of keywords to max. 5.
- Remove text highlight colour.
- Remove "data not shown" (p.36).
- Add callouts for Fig 5E, Fig 6F, Fig EV1A and Fig EV3A.
- In M&M, include a statement that informed consent was obtained from all human subjects and that the experiments conformed to the principles set out in the WMA Declaration of Helsinki and the Department of Health and Human Services Belmont Report.
- In M&M, the statistical paragraph should reflect all information that you have filled in the Authors Checklist, especially regarding randomization, blinding, replication.
- Indicate in legends exact n= and exact p= values, not a range, along with the statistical test used. To keep the figures "clear" some authors found providing an Appendix table Sx with all exact p-values preferable. You are welcome to do this if you want to.
- In addition to the accession number please provide URL for deposited datasets. Please be aware that all datasets should be made freely available upon acceptance, without restriction. Use the following format to report the accession number of your data:

[data type]: [full name of the resource] [accession number/identifier] ([doi or URL or identifiers.org/DATABASE:ACCESSION])

Please check "Author Guidelines" for more information.

<https://www.embopress.org/page/journal/17574684/authorguide#availabilityofpublishedmaterial>

2) Tables: Please rename 5 appendix tables to Table EV1 etc. Also, remove appendix tables from the table of content in the "Appendix".

3) As part of the EMBO Publications transparent editorial process initiative (see our Editorial at <http://embomolmed.embopress.org/content/2/9/329>), EMBO Molecular Medicine will publish online a

Review Process File (RPF) to accompany accepted manuscripts. This file will be published in conjunction with your paper and will include the anonymous referee reports, your point-by-point response and all pertinent correspondence relating to the manuscript. Let us know whether you agree with the publication of the RPF and as here, if you want to remove or not any figures from it prior to publication. Please note that the Authors checklist will be published at the end of the RPF. 4) Please provide a point-by-point letter INCLUDING my comments as well as the reviewer's reports and your detailed responses (as Word file).

I look forward to reading a new revised version of your manuscript.

Yours sincerely,

Zeljko Durdevic

***** Reviewer's comments *****

Referee #1 (Comments on Novelty/Model System for Author):

As mentioned in my comments it would be good to use another protocol for generating patient-derived COs (either the intrinsic one that the authors use already in their crisp-edited COs or the dorsal/ventral COs instead of the FOs)

Referee #1 (Remarks for Author):

Comments to the revised manuscript

In this manuscript Steinberg et al model with brain organoids the early infantile WWOXrelated epileptic encephalopathy and investigate the role of WWOX in embryonic human brain development. Following the previous suggestion, the authors substantially improved their manuscript and present new data. However, some points still need further attention. At least more discussion needs to be done to address all the comments raised before and now. These concerns and further suggestions for improving this work are listed below.

Major:

1. Figure 1F there is huge variability in the quantification. It is not clear how the authors find these data as statistically different. Also, there is no information in the legend on what is depicted in the plots (mean {plus minus} SEM?)

2. The figure in the response to comment 3 is blurry. I cannot read the x-axis and thus I cannot assess what the authors claim. Besides, why these data are not at minimum discussed in the manuscript? I feel that this is an important point that all the readers - not only reviewers - need to know.
3. Regarding major comment 4 (also comment raised by another reviewer). I understand that there is variability in the phenotype of the patients regarding the size of their brain however I believe that the authors should check whether this is the case in their CO cultures. The measure of the size of the organoids is not a demanding experiment and would help to clarify this point.
4. Fig. EV2 A,B show reduced SOX2 and NEUN expression. This is inconsistent. Statistical analysis needs to be done since these data (also mentioned in the previous comments) are inconsistent.
5. Regarding the way the authors addressed comment 6: I agree with the authors that the experiment they did was inconclusive and I understand that technical difficulties may arise in the electroporation of the organoids. Thus, the authors should not make any conclusion about the reason why they see increased astrocytes. In light of this, the conclusion "These findings imply that the initial increase in astrocytic markers in WWOX-depleted COs is likely due to enhanced differentiating RGs rather than proliferating APCs" is not supported. The authors should avoid any overinterpretation of their results overall in the manuscript. Besides, why in the figure presented for reviewers only the quantification of the S100b cells is not changed while they show in the manuscript the opposite (Fig.3D,F)? Another point regarding the BrdU experiment: why the authors chose to do it at late time points (it is not specified when the author did the experiment but I assume that they did it in late time points since ctrl COs have elevated S100b expression)? If this is correct, I would assume that if there is a change in the differentiation of vRGs this could be in earlier time points. Could this explain the lack of statistical difference the authors observe?
6. The caspase-3 experiment is very interesting. Quantification is however missing. For the pictures presented in Fig. EV2G it is not possible to appreciate the reduction. Please include quantification and higher magnifications.
7. Statistical analysis is missing in Fig.4E
8. Fig S2F authors claim that FOs are formed properly using SOX2 as a marker which however is not specific for the forebrain. In addition, they reply to reviewer 3 comments 12 saying that they use PAX6, however, they do not do that in organoids from family VSM. This is an important point that needs to be addressed.
9. Comment 9 was not addressed at all. The authors do not explain why they used FO protocol and not the dorsal/ventral protocols. This was an important point concerning experimental design. The authors should at minimum discuss this point if not do part of the analysis as suggested.

Minor:

1. In many places figure citation is wrong. e.g. In the first paragraph of the results, the authors mention: "Immunoblot analysis was used to assess WWOX expression in these lines (Appendix Fig S1A)." This is incorrect. These data are shown in Fig. EV1. Another example: " The mean spectral power of field recordings further showed an overall increase in power of the KO COs in the 0.25-1 Hz, typically labelled as slow wave oscillations (SWO, <1 Hz) (Fig 2B) and a decrease in the 30-79.9 high-frequency range (Fig EV1G-H-C)", Fig. EV1C shows something else. The authors should carefully check the whole manuscript for other inconsistencies.
2. Why there are supplementary figures named as Figures EV and others as Appendix Fig.S? This is very confusing. Except for the 6 main figures, the authors should have all the rest of the figures under one category.
3. Some figures are not cited in the order they are presented. e.g. EV2A,B is mentioned before EV1F-J, Fig.2E before Fig.2D ect. Authors should carefully check the whole manuscript in light of this comment and correct/rearrange the figures.

Referee #2 (Comments on Novelty/Model System for Author):

After revision, the data quality has improved and figures are presented with statistical analysis.

Referee #2 (Remarks for Author):

Although revision has improved the manuscript, below are additional comments.

1. Authors performed karyotypes and teratoma assay. They should include the data as supplemental data. Currently, the data is not included in any part of manuscript.

2. Please give description of the mutation of K01 and K02 in manuscript. It is hard to see the sequencing results in supplemental figures, Authors may show the diagram how the sequences were changed below the Sanger sequencing data.

Referee #3 (Remarks for Author):

The authors did a huge amount of work to reply to all the issues raised by the reviewers. Thus, I am supporting the publication of the manuscript in its revised form.

Point-by-point

Editor's Comments

Thank you for the submission of your revised manuscript to EMBO Molecular Medicine. We have now heard back from the three referees who we asked to re-evaluate your manuscript. As you will see from the reports below, the referees are overall supporting publication of your manuscript but also raise some concerns that should be addressed in an additional and final round of revision. Please address all the points raised by the referees and implement all suggested adjustments. We would like to encourage you to address referee #1 points 3 and 9 experimentally, however, this is not essential and can be also addressed in writing. Furthermore, we noticed that Fig EV5D contains a boxplot with 'WPM F2: n=1'; please revise and add an additional sample or consider an alternative way to visualize the quantification.

Response: *We thank the Editor for his positive evaluation of our manuscript. Per request of the Editor, we addressed all the concerns raised by the referees as explained in the point-by-point document. As for Fig EV5D, we combined F2 and M3 together (referred to as WPM P) as they both of the same genotype in relation to WWOX.*

In addition, please amend the following:

1) In the main manuscript file, please do the following:

- Correct/answer the track changes suggested by our data editors by working from the attached/uploaded document.

Response: Done.

- Reduce number of keywords to max. 5.

Response: Done.

- Remove text highlight colour.

Response: Done.

- Remove "data not shown" (p.36).

Response: Done.

- Add callouts for Fig 5E, Fig 6F, Fig EV1A and Fig EV3A.

Response: Done.

- In M&M, include a statement that informed consent was obtained from all human subjects and that the experiments conformed to the principles set out in the WMA Declaration of Helsinki and the Department of Health and Human Services Belmont Report.

Response: Done.

- In M&M, the statistical paragraph should reflect all information that you have filled in the Authors Checklist, especially regarding randomization, blinding, replication.

Response: Done.

- Indicate in legends exact n= and exact p= values, not a range, along with the statistical test used. To keep the figures "clear" some authors found providing an Appendix table Sx with all exact p-values preferable. You are welcome to do this if you want to.

Response: Done.

- In addition to the accession number please provide URL for deposited datasets. Please be aware that all datasets should be made freely available upon acceptance, without restriction. Use the following format to report the accession number of your data:

The datasets produced in this study are available in the following databases:
[data type]: [full name of the resource] [accession number/identifier] ([doi or URL
or identifiers.org/DATABASE:ACCESSION])

Please check "Author Guidelines" for more information. <https://www.embopress.org/page/journal/17574684/authorguide#availabilityofpublishedmaterial>

Response: Done.

2) Tables: Please rename 5 appendix tables to Table EV1 etc. Also, remove appendix tables from the table of content in the "Appendix".

Response: Done.

3) As part of the EMBO Publications transparent editorial process initiative (see our Editorial at <http://embomolmed.embopress.org/content/2/9/329>), EMBO Molecular Medicine will publish online a Review Process File (RPF) to accompany accepted manuscripts. This file will be published in conjunction with your paper and will include the anonymous referee reports, your point-by-point response and all pertinent correspondence relating to the manuscript. Let us know whether you agree with the publication of the RPF and as here, if you want to remove or not any figures from it prior to publication. Please note that the Authors checklist will be published at the end of the RPF.

Response: We agree to publish the RPF. Please remove any figures prior to publication.

***** Reviewer's comments *****

Referee #1 (Comments on Novelty/Model System for Author):

As mentioned in my comments it would be good to use another protocol for generating patient-derived COs (either the intrinsic one that the authors use already in their crisp-edited COs or the dorsal/ventral COs instead of the FOs)

Referee #1 (Remarks for Author):

Comments to the revised manuscript

In this manuscript Steinberg et al model with brain organoids the early infantile WWOX-related epileptic encephalopathy and investigate the role of WWOX in embryonic human brain development. Following the previous suggestion, the authors substantially improved their manuscript and present new data. However, some points still need further attention. At least more discussion needs to be done to address all the comments raised before and now. These concerns and further suggestions for improving this work are listed below.

Response: *We very much appreciate the statement of the Reviewer and thank him/her for acknowledging the significance of our paper. We hope we successfully addressed all the proposed suggestions as described below and in our revised manuscript.*

Major:

1. Figure 1F there is huge variability in the quantification. It is not clear how the authors find these data as statistically different. Also, there is no information in the legend on what is depicted in the plots (mean {plus minus}SEM?)

Response: *In order to address the variability, the ROUT test was performed (Q=1%) which identified a total of 11 outliers in all the groups (WT, KO, W-AAV). The outliers were removed, and the data was replotted, hopefully showing more uniform results. The boxplot represents the 1st and 3rd quartile, with its whiskers showing the minimum and maximum points and a central band representing the median. This description was added to the legend in an attempt to prevent any further confusion.*

2. The figure in the response to comment 3 is blurry. I cannot read the x-axis and thus I cannot assess what the authors claim. Besides, why these data are not at minimum discussed in the manuscript? I feel that this is an important point that all the readers - not only reviewers - need to know.

Response: *As requested, this figure now appears in Appendix figure S3C, and is referred to in page 14, end of 1st paragraph.*

3. Regarding major comment 4 (also comment raised by another reviewer). I understand that there is variability in the phenotype of the patients regarding the size of their brain however I believe that the authors should check whether this is the case in their CO cultures. The measure of the size of the organoids is not a demanding experiment and would help to clarify this point.

Response: *As requested, we now conducted size measurement of CO and present this in Appendix fig S2A. We didn't find a statistical significant difference between the different COs. A statement referring to this finding was added in page 6, beginning of the 2nd paragraph.*

4. Fig. EV2 A,B show reduced SOX2 and NEUN expression. This is inconsistent. Statistical analysis needs to be done since these data (also mentioned in the previous comments) are inconsistent.

Response: *We kindly thank the reviewer for this comment as it can in fact be a source of confusion – in the corrected version of the manuscript, the quantification presented in fig EV2B now also contains SEM – together with the blot presented in fig EV2A, we believe the reader will be able to notice the high heterogeneity of brain organoids (which is known in the literature). Although there is a difference between WT and KO COs, it is not much different from the differences between 2 KO COs, and therefore was considered as a random result (and not as a biological difference).*

5. Regarding the way the authors addressed comment 6: I agree with the authors that the experiment they did was inconclusive and I understand that technical difficulties may arise in the electroporation of the organoids. Thus, the authors should not make any conclusion about the reason why they see increased astrocytes. In light of this, the conclusion "These findings imply that the initial increase in astrocytic markers in WWOX-depleted COs is likely due to enhanced differentiating RGs rather than proliferating APCs" is not supported. The authors should avoid any overinterpretation of their results overall in the manuscript. Besides, why in the figure presented for reviewers only the quantification of the S100b cells is not changed while they show in the manuscript the opposite (Fig.3D,F)? Another point regarding the BrdU experiment: why the authors chose to do it at late time points (it is not specified when the author did the experiment but I assume that they did it in late time points since ctrl COs have elevated S100b expression)? If this is correct, I would assume that if there is a change in the differentiation of vRGs this could be in earlier time points. Could this explain the lack of statistical difference the authors observe?

Response: The reviewer raised a few concerns here, which will be addressed in order:

1. The claim regarding the differentiation of RGs vs proliferation of APCs – *As our experiment was inconclusive, we agree toning-down our statement is in order, therefore it was removed from the manuscript. The original methodological approach for this experiment was based on a similar experiment conducted by Blair et al., (Nature medicine, 2018), who co-stained Ki-67 with S100 β to address the source of the increased glial cells.*
2. Time point selection - *In the figure attached for the reviewers, and is not shown in the paper itself, we used BrdU-labeling to try and track the source of glial cells. This approach was unsuccessful. Either way, the labeling was preformed on week 4 organoids for 48hr, and the staining was preformed 21 days after. Generally, week 4 is a time point in which almost no astrocytes are observed, therefore we assumed mainly SOX2+ cells will be labelled. In the following 21 days, we expect some astrocytes to appear, with a bit of variability between organoids from the same batch. To our surprise,*

more astrocytes than expected appeared in the WT organoids, in contrast to our previous 4 batches.

- 3. Differences in the number of astrocytes between this figure and the manuscript – in continuation to what said regarding the time points – to our surprise, WT organoids in this batch contained more astrocytes than expected, yet, overall, the KO organoids contained more. In order to properly determine whether there is an increase in the proportions of BrdU+/S100β+ cells, we wanted to take out of the equation regional changes. In other words – if we would have quantified the astrocytes-poor areas in the WT organoids, we wouldn't have known whether the number of BrdU+/S100β+ cells is a result from the over-all low number of S100β+ cells, or from less differentiation of BrdU+ cells to astrocytes. By quantifying the same number of S100β+ cells, we hoped to get a result that reflects only the change in BrdU+/S100β+ cells.*

6. The caspase-3 experiment is very interesting. Quantification is however missing. For the pictures presented in Fig. EV2G it is not possible to appreciate the reduction. Please include quantification and higher magnifications.

Response: *As requested, a quantification is now added to the manuscript in fig. EV2H, referred to in page 11, end of 1st paragraph.*

7. Statistical analysis is missing in Fig.4E

Response: *As requested, a statistical analysis is now added in the revised version of the manuscript. Quantification and SEM appear at the bottom of Fig 4E.*

8. Fig S2F authors claim that FOs are formed properly using SOX2 as a marker which however is not specific for the forebrain. In addition, they reply to reviewer 3 comments 12 saying that they use PAX6, however, they do not do that in organoids from family VSM. This is an important point that needs to be addressed.

Response: *The requested figure is now shown in the revised manuscript under the new Appendix fig S4F.*

9. Comment 9 was not addressed at all. The authors do not explain why they used FO protocol and not the dorsal/ventral protocols. This was an important point concerning experimental design. The authors should at minimum discuss this point if not do part of the analysis as suggested.

Response: *As stated in our initial submission, most of the phenotype was observed in the cortical part of our COs – for example, the changes in the cortical layers (Fig 4F-H, EV3D-F), as stated in page 16, last paragraph. This is further supported by 1) the literature showing that the CO protocol yields mainly dorsal regions, 2) our RNA-seq data (Appendix Fig S3C) and 3) by a recent paper from our group in mice (Repudi et al., Brain, 2021). Therefore, we attempted to use the FO protocol developed by the Ming lab (Qian et al., Cell, 2016) to further test the phenotype observed upon WWOX ablation.*

The reviewer's request ("In my opinion the authors should include at least part of the analysis in the unpattern organoids and/or in ventral and dorsal assembloids/fused organoids"), we repeated our experiments in a new batch of unpatterned WSM COs (fig 5, 6 and Appendix fig S4F), which yielded similar results to the FOs, further supporting the importance of the cortex to the phenotype. We believe that our results reproduced in two different protocols and different systems (iPSCs and manipulated ESCs) support the conclusions we drew. Future work could involve additional protocols.

Minor:

1. In many places figure citation is wrong. e.g. In the first paragraph of the results, the authors mention: "Immunoblot analysis was used to assess WWOX expression in these lines (Appendix Fig S1A)." This is incorrect. These data are shown in Fig. EV1. Another example: " The mean spectral power of field recordings further showed an overall increase in power of the KO COs in the 0.25-1 Hz, typically labelled as slow wave oscillations (SWO, <1 Hz) (Fig 2B) and a decrease in the 30-79.9 high-frequency range (Fig EV1G-H-C)", Fig. EV1C shows something else. The authors should carefully check the whole manuscript for other inconsistencies.

Response: *We thank the reviewer for the keen eye. The mistakes mentioned above are now corrected, and the manuscript was carefully examined for any other misprints.*

2. Why there are supplementary figures named as Figures EV and others as Appendix Fig.S? This is very confusing. Except for the 6 main figures, the authors should have all the rest of the figures under one category.

Response: *We regret any confusion caused to the reviewer, but this is the requirement of the journal.*

3. Some figures are not cited in the order they are presented. e.g. EV2A,B is mentioned before EV1F-J, Fig.2E before Fig.2D ect. Authors should carefully check the whole manuscript in light of this comment and correct/rearrange the figures.

Response: *We thank the reviewer for the suggestion. As the main point of fig EV2A,B is to quantify the expression of GFAP (NeuN and SOX2 are quantified in fig 1C), we feel this blot should remain in fig EV2. Therefore, this callout is now removed.*

Referee #2 (Comments on Novelty/Model System for Author):

After revision, the data quality has improved and figures are presented with statistical analysis.

Response: *We very much appreciate the statement of the Reviewer and thank him/her for acknowledging the significance of our paper. We hope we successfully addressed all the proposed suggestions as described below and in our revised manuscript.*

Remarks for Author:

Although revision has improved the manuscript, below are additional comments.

1. Authors performed karyotypes and teratoma assay. They should include the data as supplemental data. Currently, the data is not included in any part of manuscript.

Response: *As requested, the data is now displayed as part of the new Appendix fig 1A-B, and are referred to in the manuscript at page 5, 1st paragraph.*

2. Please give description of the mutation of K01 and K02 in manuscript. It is hard to see the sequencing results in supplemental figures, Authors may show the diagram how the sequences were changed below the Sanger sequencing data.

Response: *As requested, the data is now displayed as part of the new fig EV1C and in Appendix fig 1C, and are referred to in the manuscript at page 5.*

Referee #3 (Remarks for Author):

The authors did a huge amount of work to reply to all the issues raised by the reviewers. Thus, I am supporting the publication of the manuscript in its revised form.

Response: *We very much appreciate the statement of the Reviewer and thank him/her for acknowledging the significance of our paper.*

4th Jun 2021

Dear Prof. Aqeilan,

We are pleased to inform you that your manuscript is accepted for publication and is now being sent to our publisher to be included in the next available issue of EMBO Molecular Medicine.

Corresponding Author Name: Rami I. Aqeilan

Manuscript Number: EMM-2020-13610